# Mapping the O-glycoproteome using site-specific extraction of O-linked glycopeptides (EXoO)

Weiming Yang ⓘD, Minghui Ao, Yingwei Hu, Qing Kay Li & Hui Zhang*

## Abstract

Protein glycosylation is one of the most abundant post-translational modifications. However, detailed analysis of O-linked glycosylation, a major type of protein glycosylation, has been severely impeded by the scarcity of suitable methodologies. Here, a chemoenzymatic method is introduced for the site-specific extraction of O-linked glycopeptides (EXoO), which enabled the mapping of over 3,000 O-linked glycosylation sites and definition of their glycans on over 1,000 proteins in human kidney tissues, T cells, and serum. This large-scale localization of O-linked glycosylation sites demonstrated that EXoO is an effective method for defining the site-specific O-linked glycoproteome in different types of sample. Detailed structural analysis of the sites identified revealed conserved motifs and topological orientations facing extracellular space, the cell surface, the lumen of the Golgi, and the endoplasmic reticulum (ER). EXoO was also able to reveal significant differences in the O-linked glycoproteome of tumor and normal kidney tissues pointing to its broader use in clinical diagnostics and therapeutics.

**Keywords** glycoproteomics; glycosylation; O-GalNAc; O-linked; site-specific
**Subject Categories** Genome-Scale & Integrative Biology; Methods & Resources; Post-translational Modifications, Proteolysis & Proteomics
**Mol Syst Biol. (2018) 14: e8486**

## Introduction

Protein glycosylation is arguably the most diverse and sophisticated form of protein modification which drastically escalates protein heterogeneity to facilitate functional plasticity (Varki, 2011; Tran & Ten Hagen, 2013). Compared to N-linked glycosylation, the study of O-linked glycosylation has proved difficult due to the structural complexity of the glycans added and the technical challenges posed to definitively characterizing them (Jensen *et al*, 2010; Qin *et al*, 2017; Darula & Medzihradszky, 2018). The process of O-linked glycosylation attaches different O-linked glycans to Ser and Thr residues, or less commonly Tyr residues in the human proteome (Halim *et al*, 2011; Trinidad *et al*, 2013). Among the different types of O-linked glycosylation seen, O-linked N-acetyl-galactosamine

(O-GalNAc) addition is a major type (Jensen *et al*, 2010; Chia *et al*, 2016; Darula & Medzihradszky, 2018). Definitive characterization of O-linked glycoproteins requires quantitative analysis of O-linked glycosylation sites and their corresponding glycans. In contrast to N-linked glycosylation, where a consensus glycosylation motif has been identified, there is no consensus O-linked glycosylation motif for the amino acid residues surrounding the glycosylated Ser or Thr (Nishikawa *et al*, 2010; Darula & Medzihradszky, 2018). The cellular machinery for O-linked glycosylation, located primarily in the Golgi apparatus, is believed to operate stochastically in response to changes in a wide range of both intrinsic and extrinsic factors (Chia *et al*, 2014; Bard & Chia, 2016). The presence of up to 20 GalNAc-transferases (GalNAc-Ts) for adding the initial sugar to amino acid residues in different sequence regions further complicates the dynamic regulation of O-linked glycosylation (Bennett *et al*, 2012). As a consequence, O-linked glycosylation can exhibit high heterogeneity in different cells, tissues, and diseases (Steentoft *et al*, 2013; Medzihradszky *et al*, 2015).

The enrichment of O-linked glycopeptides from complex biological samples is an essential prerequisite to definitive identification of the otherwise low abundance O-linked glycoproteome (Woo *et al*, 2015). To this end, a range of different enrichment methodologies have been reported including the use of lectins (Darula & Medzihradszky, 2009; King *et al*, 2017), HILIC (Hagglund *et al*, 2007; Yang *et al*, 2017), hydrazide chemistry (Nilsson *et al*, 2009; Halim *et al*, 2012), metabolic labeling (Woo *et al*, 2015, 2017), and a gene-engineered cell system named "SimpleCell" (Steentoft *et al*, 2011). The enrichment methodologies have proved to be useful to study O-linked glycoproteome in different biological systems. In addition, the precise pinpointing of O-linked glycosylation sites and their corresponding site-specific glycans has proved to be even more challenging. Mass spectrometry (MS) using electron transfer dissociation (ETD) for fragmentation has to date been the prevailing analytical tool to localize O-linked glycosylation sites (Zhu *et al*, 2013). The site localization using ETD is efficient for mapping glycosylation sites when the glycosylation sites are located in peptides with high charge states and the presence of convincing peptide fragment ions covering the O-linked glycosylation sites (Good *et al*, 2007; Darula *et al*, 2012; Mulagapati *et al*, 2017). In the largest O-glycoproteome study, lectin enrichments of desialylated core 1 O-linked glycopeptides from plasma using PNA and VVA and ETD-MS2 analysis, 354 unique O-linked glycosylation sites were reported

Department of Pathology, Johns Hopkins University School of Medicine, Baltimore, MD, USA
*Corresponding author. Tel: +1 410 502 8149: E-mail: huizhang@jhu.edu

among the total of 1,123 O-glycosites identified from plasma, platelets, and endothelial cells (King *et al*, 2017). Alternative to ETD fragmentation, HCD-MS2 may provide an efficient fragmentation method to identify glycopeptides, but the O-linked glycosylation sites cannot be confidently assigned due to preferential fragmentation of O-linked glycans during HCD mode (Yang *et al*, 2014). Alternatively, chemical deglycosylation such as β-elimination Michael addition can be used to substitute O-linked glycans for glycosylation site identification (Nika *et al*, 2013). However, a variety of complications and non-specific substitutions at other post-translational modifications or at unmodified Ser and Thr limits the performance of this alternative method when employed on complex samples (Nika *et al*, 2013). In sharp contrast, the identification of N-linked glycosylation sites and N-linked glycans is highly efficient making use of a specific enzyme (PNGase F) and the consensus NXS/T motif (X ≠ Pro; Bause, 1983). PNGase F simultaneously liberates N-linked glycans and converts the Asn in the glycosylation site to Asp (deamination) thereby identifying the site (Zhang *et al*, 2003; Sun *et al*, 2016). Unfortunately, there is no PNGase F equivalent for releasing O-linked glycans and localization of O-linked glycosylation sites. Consequently, any new methodology capable of large-scale analysis of site-specific O-linked glycoproteome would be highly advantageous.

Here, a new chemoenzymatic method is introduced for extraction of O-linked glycopeptides, named EXoO. It has been designed to simultaneously enrich and identify O-linked glycosylation sites and define their site-specific glycans with core 1 structures with or without sialic acids. EXoO comprises four steps: (i) digestion of protein samples to generate peptides, (ii) conjugation of peptides to a solid support, (iii) release of intact O-linked glycopeptides at O-linked glycosylation sites using an O-linked glycan-dependent endo-protease named OpeRATOR, (iv) analysis of the released intact O-linked glycopeptides by LC-MS/MS. OpeRATOR, identified from the mucin degrading human intestinal bacterium *Akkermansia muciniphila*, recognizes O-linked glycans and cleaves O-linked glycopeptides at the N-termini of O-linked glycan-occupied Ser or Thr to release site-specific O-linked glycopeptides with their conjugated O-linked glycans. During the peer-review period of our study, a manuscript by Yang *et al* (2018), described the analysis of O-linked glycosylation sites from several simple glycoproteins including fetuin, mucin, and Zika viral proteins. This study took use of the enzyme, OpeRATOR, with cleavage of peptide sequences at the N-termini of the O-linked glycosylation sites (Yang *et al*, 2018). However, the glycan specificity and cleavage specificity for the O-linked glycosylation sites by OpeRATOR enzyme are not clearly defined. In this study, method EXoO was developed using bovine fetuin and then benchmarked to identify a plethora of O-linked glycosylation sites using human kidney tissues, serum, and T cells before being applied to defining the aberrant O-linked glycoproteome in human kidney tumor tissue.

# Results

### Extraction of site-specific O-linked glycopeptides

In EXoO, proteins are first digested to generate peptides, which are then conjugated to a solid support. After washing, the O-linked glycopeptides are enzymatically released from the support using an endo-protease OpeRATOR that requires the presence of O-linked glycans to specifically cleave on the N-terminal side of O-linked glycan-occupied Ser or Thr (Fig 1A). To demonstrate proof of principle, bovine fetuin was analyzed and the six known O-linked glycosylation sites documented in the UniProt database were pinpointed at Ser-271, Thr-280, Ser-282, Ser-296, Thr-334, and Ser-341 (Dataset EV1). In addition, a new O-linked glycosylation site at Ser-290 was also identified (Dataset EV1 and Appendix Fig S1). Of note, O-linked glycans were still attached to the site-specific O-linked glycopeptides as confirmed by the detection of oxonium, peptide (Y0), and less commonly identified peptide + HexNAc (Y1) ions in the MS/MS spectrum (Fig 1B). The detection of oxonium ions in the MS/MS spectrum is particularly useful for obtaining the correct identification of O-linked glycopeptides. In addition, the chemical conjugation of peptides to a solid support allows efficient washing and specific enzymatic release of intact O-linked glycopeptides. As a result, 193 peptide spectrum matches (PSMs) were assigned to fetuin site-specific O-linked glycopeptides with Ser or Thr at the N-termini of peptides, glycan modification, and oxonium ions in the MS/MS spectra from a total of 270 assigned PSMs, indicating a specificity of approximately 71.5% for O-linked glycopeptide enrichment using EXoO (Dataset EV1). The analysis of fetuin demonstrated the ability of EXoO to enrich and identify O-linked glycopeptides at specific O-linked glycosylation sites and their corresponding O-linked glycans.

### Large-scale and precision mapping of the O-linked glycoproteome in human kidney tissues, T cells, and serum sample

EXoO was benchmarked using human kidney tissue, T cells, and serum to determine performance of the method in samples with differing levels of protein complexity. To do this, O-linked glycopeptides were extracted using EXoO and fractionated into 24 fractions and then subjected to LC-MS/MS analysis (Fig 2A). To study kidney tissue, paired tumor and normal tissues were collected from three patients. The extracted proteins from these tissues were separately pooled to generate two samples, that is, tumor and normal. After analysis with 1% false discovery rate (FDR) at PSM level, 35,848 PSMs were assigned to 2,804 O-linked glycopeptides containing 1,781 O-linked glycosylation sites from 592 glycoproteins (Dataset EV2). A number of 112 spectra with different sequences, charge, peptide length, scores, and glycan compositions were annotated (Appendix Fig S2). When the EXoO approach was applied to the analysis of T cells, 4,623 PSMs were assigned to contain 1,295 O-linked glycosylation sites from 1,982 O-linked glycopeptides and 590 glycoproteins (Dataset EV3). Finally, we studied human serum that contains a number of highly glycosylated proteins and has been previously subjected to detailed mapping of N-linked glycosylation sites and N-linked glycans but for which there has been little success in mapping of O-linked glycosylation sites and O-linked glycans (Zhang *et al*, 2005; Stumpo & Reinhold, 2010; Yabu *et al*, 2014; Darula *et al*, 2016; Hoffmann *et al*, 2016). With 1% FDR, 6,157 PSMs were assigned to 1,060 O-linked glycopeptides with 732 O-linked glycosylation sites from 306 glycoproteins being identified (Dataset EV4). This analysis of human tissue, T cells, and serum demonstrated that EXoO is a highly effective tool for accessing the O-linked glycoproteome in different types of samples.

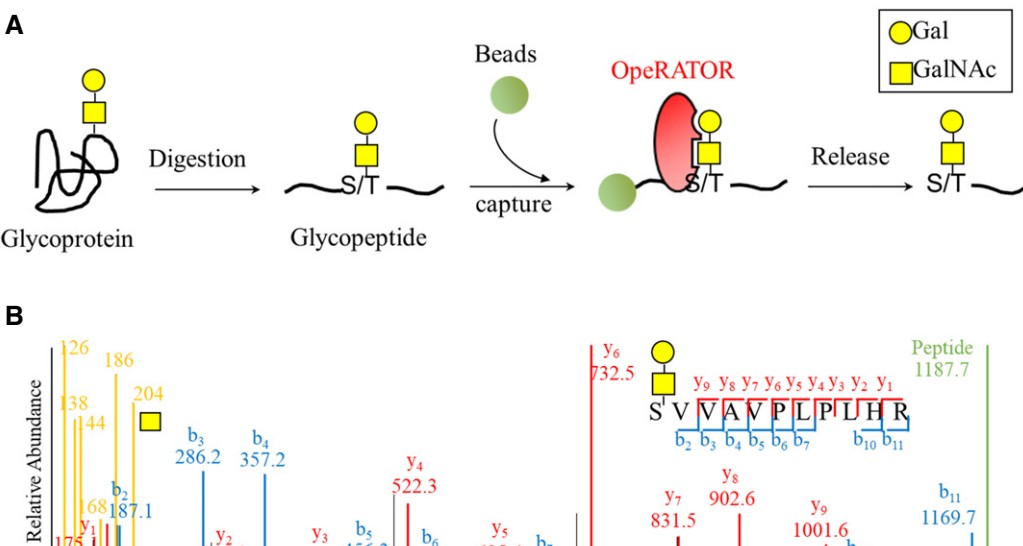

**Figure 1.  EXoO procedure for mapping the site-specific O-linked glycoproteome.**
A  Schematic of EXoO process for precision mapping of O-linked glycosylation sites and site-specific glycans.
B  MS/MS spectrum of the site-specific O-linked glycopeptide at Ser-296 in bovine fetuin.

## Specificities of OpeRATOR for peptides and O-linked glycans

To define cleavage specificities of peptides and O-linked glycans of OpeRATOR in the complex samples, sequential ETD/HCD-MS2 analysis was conducted on serum O-linked glycopeptides generated using the EXoO method. With 1% FDR, HCD-MS2 and ETD-MS2 identified 85 and 40 unique intact glycopeptides, respectively, with 27 glycopeptides identified in both modes (Dataset EV5). Next, the ambiguity of O-linked glycosylation site at the first amino acid position on glycopeptides was determined in the PSMs generated by ETD-MS2. Among the 113 PSMs from ETD-MS2 analysis, 105 PSMs were assigned to have core 1 O-linked glycans conjugated at glycosylation sites localized at the first amino acid position of glycopeptides with ptmRS site probabilities of over 99% (Dataset EV5). O-linked glycosylation sites with core 1 O-linked glycans in seven PSMs could not be located by ETD spectra (Dataset EV5). One PSM was identified to have the O-linked glycosylation site assigned at the sixth amino acid position on a serine (Dataset EV5). However, a second PSM of the same precursor was assigned to the first threonine residue (Dataset EV5). The observation of two PSMs of the same precursor with different site localization suggested that the site localization for this glycopeptide might not be confident for site assignment. Therefore, ETD-MS2 provided that a cleavage specificity by OpeRATOR was at the N-termini of the O-linked glycosylation sites with core 1 glycans. HCD-MS2 appeared to identify more unique glycopeptides compared to ETD-MS2 that might be due to shorter glycopeptides with low charge states generated by trypsin and OpeRATOR digestion, while glycopeptides identified by ETD-MS2 contained only +3 charge and above (Dataset EV5). Therefore, one of the advantages of EXoO method for O-linked glycoproteomics analysis empowered efficient O-linked glycosylation site localization by high cleavage specificity of OpeRATOR that was confirmed by ETD-MS2.

The precise specificity of OpeRATOR for different O-linked glycans remains unclear. Analysis of our data from tissue, serum, and cells revealed that approximately 69% of total PSM contained glycan composition Hex(1)HexNAc(1) that was most likely to be core 1 mucin-type glycan Gal-GalNAc. Therefore, it is possible to define that the O-linked glycopeptide contained Hex(1)HexNAc(1) or most likely to be Gal-GalNAc with or without sialic acid at the site of O-linked glycosylation. These data could also be explained as that the major glycan composition for site-specific O-linked glycopeptide is the core 1 structure Hex(1)HexNAc(1) that is prevalent in a wide range of glycoproteins from different cell types compared to the relatively restricted presence of other core structures seen in specific tissue and cell types (Brockhausen & Stanley, 2015). However, the fact that other glycoforms accounted for approximately 31% of total identified glycan compositions argues that further investigation is needed to definitively establish the glycoform specificity of OpeRATOR. In addition, the possibility of multiple glycans on a glycopeptide demands caution in the data interpretation to define site-specific glycan composition. For example, two Hex(1)HexNAc(1) on a glycopeptide could yield a glycan composition of Hex(2)HexNAc(2) in the result. EXoO may miss O-linked glycosylation sites that are not in an appropriate peptide length for identification. It can be anticipated that using enzymes other than trypsin for generating peptides will increase the identification of O-linked glycosylation sites (Choudhary *et al*, 2003).

## Characterizing the O-linked glycoproteome

Our large-scale analysis mapped 3,055 O-linked glycosylation sites from 1,060 glycoproteins in kidney tissues, T cells, and serum (Dataset EV6). To compare the EXoO identified sites to that reported previously, 2,746 reported O-GalNAc sites were

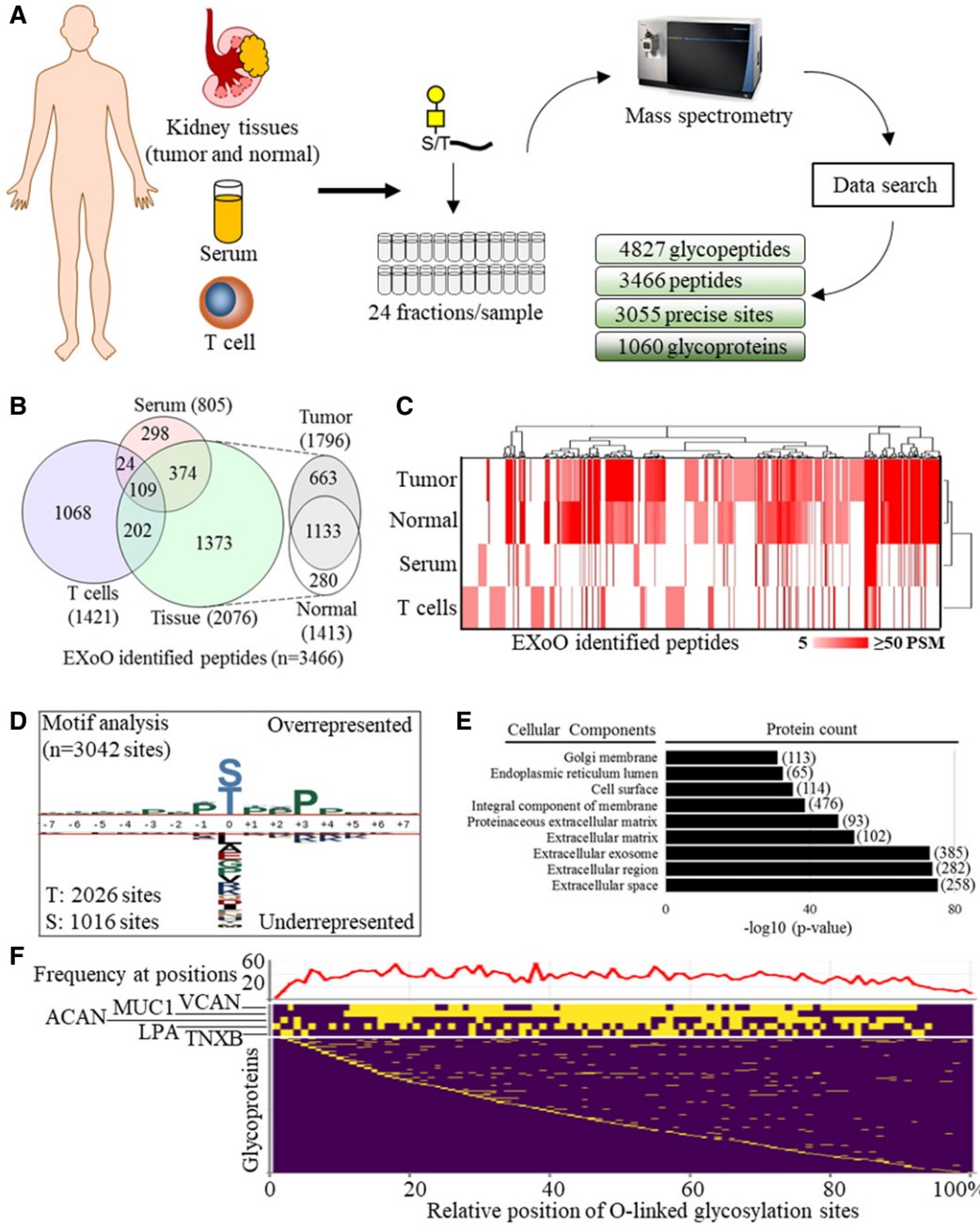

**Figure 2.  Mapping of O-linked glycoproteome.**

A   Workflow to map site-specific O-linked glycoproteome in human samples.

B   Distribution of EXoO identified peptides in different samples.

C   Unsupervised hierarchical clustering of samples and PSM number of peptides to show that EXoO identified peptides exhibited different distribution and relative abundance in samples. Euclidean distance and city block distance were used for clustering of samples and peptides, respectively.

D   Analysis of amino acid sequence surrounding O-linked glycosylation sites.

E   Cellular component analysis of O-linked glycoproteins.

F   Landscape distribution of the O-linked glycosylation sites and its frequency in proteins. Relative position was calculated using amino acid position of the sites to divide total number of amino acids of the protein and time a hundred to show in percentage. Lower panel: Relative position of the sites in proteins. Proteins with positions close to protein N-termini were ranked to the top. Middle panel: Relative position of sites in five proteins with the highest number of sites. Protein with higher number of sites was ranked to the top. Upper panel: Frequency of the sites in proteins. The *y*-axis of upper panel shows the number of sites at the relative position of proteins.

collected from O-GalNAc human SimpleCell glycoproteome DB (Steentoft *et al*, 2011, 2013), PhosphoSitePlus (Hornbeck *et al*, 2015), and UniProt database (UniProt Consortium, 2018).

Remarkably, EXoO identified 2,580 novel O-linked glycosylation sites, an approximately 94% increase in the known sites, which however are mapped primarily using engineered cell lines.

To determine sample-specific O-linked glycoproteome, the distribution of EXoO identified peptides in different samples was determined. Kidney tissue and T cells had a large number of unique peptides compared to that seen for serum, with more than half of peptides detected in serum also being identified in the tissue sample, possibly due to the presence of serum in tissue samples (Fig 2B). To visualize the relative abundance of peptides in different samples, the PSM numbers of peptides, which are suggestive of relative abundance, were clustered by unsupervised hierarchical clustering (Fig 2C). This showed that not only that the peptides differed between samples but also that their relative abundances were markedly divergent between samples (Fig 2C). Interestingly, immunoglobulin heavy constant alpha 1 (IGHA1) has the highest PSM number in the normal tissue and serum but had the second highest PSM number in the tumor tissue where versican core protein (VCAN) scored the highest PSM number suggesting their relatively high abundance for detection and aberrant O-linked glycosylation of VCAN in tumor tissue. In the case of IGHA1, four of the five known sites on Ser residues and two new sites on Thr residues were mapped supportive of EXoO's capacity to both localize known and discover new O-linked glycosylation sites. Overall, these data suggest that protein O-linked glycosylation is highly dynamic and may exhibit a disease-specific signature.

To identify possible O-linked glycosylation motifs, the amino acids ($\pm 7$ amino acids) at and surrounding 3,042 of the sites mapped in this study were analyzed. O-linked glycan addition at Thr and Ser accounted for 67.6 and 22.4% of the sites, respectively (Fig 2D). Analysis of the surrounding sequence motifs revealed that Pro was overrepresented at the $+3$ and $-1$ positions irrespective of which amino acid (Thr or Ser) was glycosylated or sample type (Fig 2D and Appendix Fig S2). Overall enrichment of Pro was observed in the amino acids surrounding O-linked glycosylation sites (Appendix Fig S2). Thirteen O-linked glycosylation sites were not used in the motif analysis because they were located close to the termini of proteins concerned and consequently did not have enough surrounding amino acids to allow for full motif analysis.

Gene ontology (GO) analysis of EXoO identified glycoproteins was carried out, and this showed that extracellular space, the cell surface, the ER lumen, and the Golgi membrane were the major cellular components for O-linked glycoproteins (Fig 2E). Analysis of biological process and molecular function suggested various activities and functionalities associated with O-linked glycoproteins, consistent with their important role in different aspects of biology (Appendix Fig S3). Specifically, extracellular matrix organization, cell adhesion, and platelet degranulation were the biological processes most represented in the glycoproteins identified (Appendix Fig S3), whereas heparin binding, calcium ion binding, and integrin binding were the top molecular functions identified (Appendix Fig S3).

To overview the positional distribution of the O-linked glycosylation sites identified, the relative position of the sites in the proteins was determined and arranged relative to the N-terminus of the glycoprotein in question (Fig 2F lower panel). In addition, frequency of the sites at the relative position of proteins was calculated (Fig 2F upper panel). It was found that the sites had relatively even distribution across the protein sequence but less frequent at protein termini (Fig 2F upper panel). Strikingly, 20 proteins were seen to contain over 20 sites. Five proteins with the highest number of sites were zoomed for clear visualization in Fig 2F middle panel.

These heavily glycosylated proteins appeared to show continuous clusters of many vincinal sites that nearly cover the whole proteins such as VCAN, mucin-1 (MUC1), and aggrecan core protein (ACAN). The cluster of sites could be relatively short while distributed evenly as seen in apolipoprotein (LPA) and Tenascin-X (TNXB). Among these heavily O-linked glycoproteins, VCAN contained the highest number of sites reaching 165 sites with distinct peptide sequences surrounding the sites, whereas MUC1 contained 161 sites, the second highest, but composited from only six distinct sequence repeats. ACAN, LPA, and TNXB were heavily O-linked glycosylated to have 82, 73, and 44 sites, respectively. Analysis of the site distribution on glycoproteins demonstrated advantage of EXoO to study heavily O-linked glycoproteins that is difficult to be analyzed by current analytical approach due to structural complexity and resistance to enzymatic digestion.

To determine localization of the sites to protein structures, protein topological and structural annotations were retrieved from UniProt database and mapped to the EXoO identified sites. It was found that approximately 28.3 and 10.3% of the sites were predicted to localize in extracellular and luminal region, respectively (Appendix Fig S4). In contrast, only approximately 1.6% of the sites were predicted in cytoplasmic compartment (Appendix Fig S4). Approximately 5% of the sites were associated with Ser/Thr/Pro-rich region but weaker correlation of the sites to other protein structures including repeats, coiled-coil, beta strand, helix, turn, and signal peptides (Appendix Fig S4). Close to none correlation of the sites to intra- and transmembrane region of proteins was observed. The structural correlation of the sites to extracellular, lumen, and Ser/Thr/Pro-rich regions coincided with the location of O-linked glycoproteins to present on extracellular space, the cell surface, the ER, and the Golgi lumen for various functionality.

## Mapping aberrant O-linked glycoproteome associated with human kidney tumor

To identify changes in the O-linked glycoproteome between normal and tumor kidney tissue, spectral counting label-free quantification of the EXoO identified peptides was used (Fig 3A). This identified 56 O-linked glycoproteins as exhibiting significant change using scoring criteria of at least a twofold change together with a difference in at least 10 PSMs between normal and tumor samples (Fig 3B and Appendix Table S1). The most striking change observed was the dramatic increase in O-linked glycans, primarily in the core 1 structure Hex(1)HexNAc(1) across the 163 and 82 sites mapped in VCAN and ACAN, respectively (Fig 3C). For example, 35 PSMs were found at Thr-2983 of VCAN from tumor tissue but none in normal tissue samples (Fig 3C red asterisk in upper panel). Similarly, 109 PSMs were detected at Thr-374 of ACAN from tumor tissue and only five in normal tissue (Fig 3C red asterisk in lower panel). Owing to unclear substrate specificity of OpeRATOR for O-linked glycans, the site-specific O-linked glycosylation by glycans in addition to Core 1 glycans merits future investigation. VCAN and ACAN are known proteoglycans that have long sugar chains in normal condition (Binder *et al*, 2017). The extensive addition of short core 1 O-linked glycans to these two proteins would be expected to enhance their mucin-type properties such as resistance to enzymatic digestion and improved stiffness, and this in turn may alter their biological and biomechanical properties producing some remodeling of the tumor

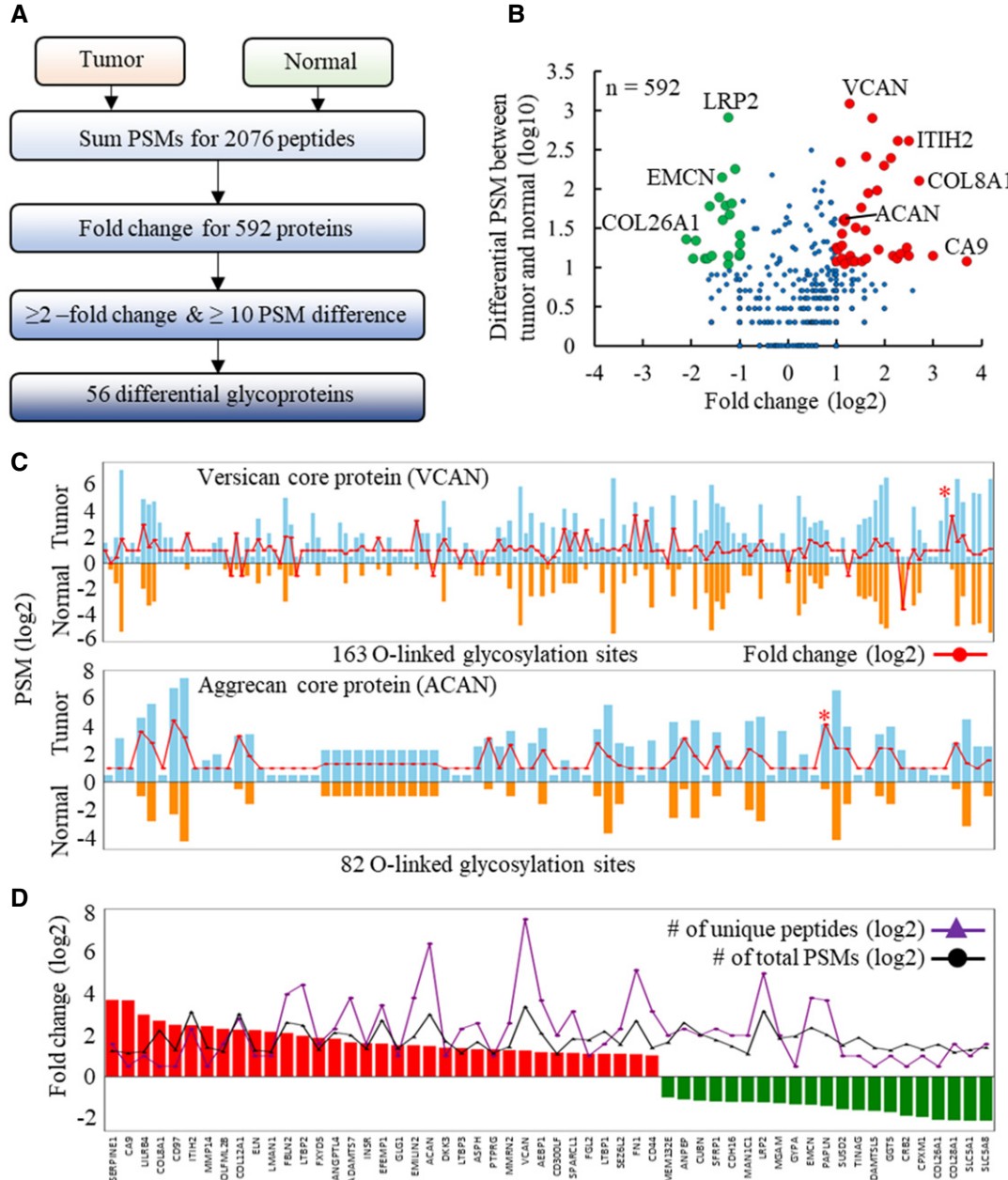

**Figure 3. Comparative analysis of the O-linked glycoproteome of normal and tumor-derived kidney tissue.**

A   Data processing steps in label-free quantification to identify differential glycoproteins between tumor and normal kidney tissues.

B   Volcano plot of differentially expressed O-linked glycoproteins between tumor and normal tissues. A total of 592 glycoproteins are plotted in the Volcano plot according to their fold change (log2) and number of differential PSM between cancer and normal (log10). Red and green dots indicate significantly upregulated and downregulated proteins, respectively.

C   Extensive addition of O-linked glycans at sites covering versican core protein (VCAN) (upper panel) and aggrecan core protein (ACAN) (lower panel). Fold change for specific site between tumor and normal was showed as connected dots in red. Sites only detected in tumor or normal were assigned two- or 0.5-fold change, respectively. Red asterisks indicate sites with the highest PSM number divergent between tumor and normal tissues.

D   The 56 O-linked glycoproteins with significant change between tumor and normal.

microenvironment (Kufe, 2009). In addition to VCAN and ACAN, an average of 4.3-fold increase was detected in 14 sites across fibulin-2 (FBLN2), a glycoprotein known to be involved in stabilizing the VCAN and ACAN network for growth and metastasis of tumor (Olin *et al*, 2001; Baird *et al*, 2013; Fig 3D and Appendix Table S1). Remodeling of the extracellular matrix (ECM) in tumor tissue might

be further underpinned by the significant changes in other ECM-related proteins such as ELN, LTBP1, LTBP2, LTBP3, EMILIN2, FN1, CDH16, and EMCN and collagens COL8A1, COL12A1, COL28A1, and COL26A1, and enzymes including ITIH2, MMP14, ADAMTS7, SERPINE1, ANPEP, PAPLN, ADAMTSL5, GGT5, and CPXM1 detected in tumor tissue (Fig 3D and Appendix Table S1).

This type of fine-tuning of the ECM network might be critical to supporting tumorigenesis and tumor progression (Lu *et al*, 2012). In addition, carbonic anhydrase 9 (CA9) and angiopoietin-related protein 4 (ANGPTL4), proteins known to respond to tumor hypoxia (Sedlakova *et al*, 2014; Carbone *et al*, 2018), showed 13- and 3.6-fold increases, respectively, in tumor tissue (Fig 3D and Appendix Table S1). Finally, EGF-containing fibulin-like extracellular matrix protein 1 (EFEMP1), which binds to epidermal growth factor receptor (EGFR) to promote tumor growth, invasion, and metastasis (Yin *et al*, 2016), showed a threefold increase across seven O-linked glycosylation sites in tumor tissue (Fig 3D and Appendix Table S1). By contrast, both IGHA1 and MUC1, known O-linked glycoproteins, showed no detectable change between normal and tumor tissue indicating that the changes in O-linked glycoproteins observed in tumor tissue are highly selective.

## Discussion

A novel tool (EXoO) has been developed for the combined mapping of O-linked glycosylation sites in proteins and the definition of the O-linked glycans at those sites. The main advantages of EXoO are (i) applicability for analysis of clinical samples including tissue, body fluid, and primary cells; (ii) precise localization of O-linked glycosylation sites; (iii) simultaneous definition of O-linked glycans at the glycosylation sites; and (iv) no requirement for ETD mass spectrometry for site localization. The effectiveness of the method derives from the specific enrichment of O-linked glycopeptides at specific glycosylation sites using the tandem action of a solid support and the O-linked glycan-specific OpeRATOR enzyme. The solid support specifically binds peptides and maximizes the removal of non-bound molecules, while the OpeRATOR enzyme specifically cleaves on the N-terminal side of glycan-occupied Ser and Thr residues of the bound peptides to release O-linked glycosylation sites at the N-terminus of peptides enabling localization of O-linked glycosylation sites. The O-linked glycan remains attached to the released O-linked glycopeptides and provides oxonium ions in the MS/MS spectrum to facilitate confident identification. As stated by the manufacturer, the use of sialidase in the procedure facilitated efficient cleavage by OpeRATOR that was used to improve mapping of O-linked glycosylation sites. The addition of sialidase could be omitted if the study focus is to define site-specific glycan structures with sialic acid.

Analysis of the more than 3,000 O-linked glycosylation sites identified by EXoO revealed many glycoproteins that were previously not known to be modified by O-linked glycosylation. Many of those identified were mucin-type glycoproteins whose mucin domain contains clusters of dense O-linked glycans that protect the underlying peptide backbone from normal proteolytic digestion, and consequently, typical proteomic analysis would not contain detailed information on many of these domains. By contrast, OpeRATOR is naturally designed to dissect such mucin-type O-glycan-rich regions. Therefore, using EXoO allowed detailed mapping of over one hundred sites on VCAN and MUC1. MUC1 has been reported to be an important molecule in many research areas including different cancers, immunity, and immunotherapy (Hanisch, 2005; Tarp & Clausen, 2008; Beatson *et al*, 2016; Hanson & Hollingsworth, 2016). The use of EXoO therefore is advantageous to reveal new biological insight regarding mucin-type glycoproteins. Motif analysis of the amino acid sequence surrounding these O-linked glycosylation sites revealed that Pro was favored at the +3 and −1 positions. This was consistent with previous reports, which gives some validation of the O-linked glycosylation sites identified using EXoO (Christlet & Veluraja, 2001; Julenius *et al*, 2005). However, achieving a better understanding the structural and functional roles of the O-linked glycans in these proteins certainly merits future investigation.

Compared to other O-linked glycoproteomic methods (Nilsson *et al*, 2009; Steentoft *et al*, 2011; Woo *et al*, 2015; Darula *et al*, 2016; Hoffmann *et al*, 2016; King *et al*, 2017; Qin *et al*, 2017), EXoO identified a large number of O-linked glycosylation sites and glycoproteins with 2,580 novel O-linked glycosylation sites that are not reported in three major database including O-GalNAc human SimpleCell glycoproteome DB (Steentoft *et al*, 2011, 2013), PhosphoSitePlus (Hornbeck *et al*, 2015), and UniProt database (UniProt Consortium T, 2018). It also identified aberrant expression of O-linked glycoproteins in kidney tumor tissue compared to normal tissue pointing to its utility in clinical investigations. Given these advantages of EXoO, it is anticipated that it will be widely applied in studies to analyze O-linked glycosylation of proteins.

## Materials and Methods

### Reagents and tools table

| Reagent/Resource | Reference or source | Identifier or catalog number |
|---|---|---|
| **Experimental models** | | |
| CEM CD4[+] cells | NIH AIDS reagent program | 117 |
| Kidney tumor and normal tissues | This study | |
| **Chemicals, enzymes and other reagents** | | |
| Human serum | Sigma Aldrich | H4522-100ML |
| Heat inactivated fetal bovine serum (HI-FBS) | ThermoFisher Scientific | 10082147 |
| OpeRATOR | Genovis | G2-OP1-020 |
| Sequencing grade modified trypsin | Promega | V5111 |

**Reagents and tools table (continued)**

| Reagent/Resource | Reference or source | Identifier or catalog number |
|---|---|---|
| Penicillin-Streptomycin | ThermoFisher Scientific | 15070063 |
| RPMI 1640 medium | ThermoFisher Scientific | 11875119 |
| Certified Sep-Pak tC18 3 cc vac cartridge | Waters | 186004619 |
| HyperSep™ retain AX cartridges | ThermoFisher Scientific | 60107-406 |
| Pierce™ BCA protein assay kit | ThermoFisher Scientific | 23227 |
| UltraPure™ dithiothreitol | ThermoFisher Scientific | 15508013 |
| Iodoacetamide | Sigma Aldrich | I1149-5G |
| Ammonium bicarbonate | Sigma Aldrich | 09830-500G |
| Urea | ThermoFisher Scientific | 29700 |
| Ammonium hydroxide solution | Sigma Aldrich | 338818-1L |
| O-Methylisourea hemisulfate salt | Sigma Aldrich | 455598-100G |
| Fetuin from fetal bovine serum | Sigma Aldrich | F3004-250MG |
| AminoLink™ plus coupling resin | ThermoFisher Scientific | 20505 |
| Sodium cyanoborohydride | Sigma Aldrich | 156159-10G |
| Trizma® hydrochloride solution | Sigma Aldrich | T2663-1L |
| NaCl Solution, 5 M | Santa Cruz Biotechnology | sc-295833 |
| Trifluoroacetic acid | Sigma Aldrich | 302031-100ML |
| Acetonitrile, Optima™ LC/MS grade | ThermoFisher Scientific | A955-4 |
| Triethylammonium acetate buffer | Sigma Aldrich | 90358-100ML |
| Pierce™ formic acid, LC-MS grade | ThermoFisher Scientific | 85178 |
| **Software** | | |
| Proteome Discoverer™ software | ThermoFisher Scientific | |
| Trans-proteomic pipeline | Institute for Systems Biology | |
| | Deutsch et al (2015) | |
| Online software pLogo | University of Connecticut | |
| | O'Shea et al (2013) | |
| Qcanvas software | Center for Advanced Bioinformatics & Systems Medicine | |
| | Kim et al (2012) | |
| Online software DAVID informatics | Laboratory of Human Retrovirology and Immunoinformatics (LHRI) | |
| | Huang da et al (2009) | |
| Online software UniProt | https://www.uniprot.org/ | |

## Methods and Protocols

### Solid-phase extraction of site-specific O-linked glycopeptides from fetuin

Bovine fetuin (P12763) was denatured in buffer containing 8 M urea and 1 M ammonium bicarbonate (AB) and reduced in 5 mM DTT at 37°C for 1 h. Proteins were alkylated in 10 mM iodoacetamide at room temperature (RT) for 40 min in the dark. The resulting samples were diluted eightfold using 100 mM AB buffer before adding trypsin (enzyme/protein ratio of 1/40 w/w) and incubating at 37°C for 16 h. Following digestion, peptides were desalted using a C18 column (Waters, Milford, MA) according to manufacturer's instructions.

Peptides were conjugated to AminoLink resin (Pierce, Rockford, IL) as previously described (Sun et al, 2016). Briefly, the pH of the peptide containing eluate of the C18 column was adjusted to 7.4 by adding phosphate buffer (pH 8.0). Peptides were then incubated with the resin (100 μg/100 μl resin, 50% slurry) and 50 mM sodium cyanoborohydride (NaCNBH₃) at RT for at least 4 h or overnight with shaking. The resin was then blocked by adding 1 M Tris–HCl buffer (pH 7.4) containing 50 mM NaCNBH₃ at RT for 30 min. The resin was washed three times with 50% acetonitrile, 1.5 M NaCl, and 20 mM Tris–HCl buffer (pH 6.8). O-linked glycopeptides were released from the resin by incubation with OpeRATOR™ and SialEXO™ (1 unit/1 μg peptides each enzyme, Genovis Inc, Cambridge, MA) in 100 μl of 20 mM Tris–HCl buffer (pH 6.8) at 37°C for 16 h according to manufacturer's instructions. Following this 16-h incubation, the released peptides in solution were collected, and the resin was washed twice with 20 mM Tris–HCl buffer (pH 7.4) to collect the remaining peptides.

The pooled peptides were then desalted on a C18 column and dried by lyophilization.

### Extraction of O-linked glycopeptides from human kidney tissue, serum, and T cells

Collection and use of human tissue have been approved by Johns Hopkins Institutional Review Board (IRB). Kidney tumor was categorized as being clear cell renal cell carcinomas (CCRCC), and samples of tumor tissue were stored at −80°C before use. Control normal kidney tissue samples were collected from the same individuals. Proteins from human kidney tissues, serum (Sigma-Aldrich, St. Louis, MO), and CEM T cells (NIH AIDS Reagent Program) were trypsin-digested as described above in the analysis of fetuin section. Following digestion, guanidination and desalting of peptides were conducted on a C18 column using procedure described previously to recover the Lys-containing peptides from complex samples (Nika *et al*, 2013). Briefly, peptides were loaded on a pre-conditioned C18 column. The column was then sequentially washed three times with 0.1% TFA and then guanidination solution (equal volumes of 2.85 M aqueous ammonia, 0.6 M O-methylisourea, and 0.1% TFA, final pH 10.5). After the final wash, guanidination solution was added to cover the C18 material in the column and it was incubated at 65°C for 20 min. Following this incubation, the column was cooled to RT and washed three times with 0.1% TFA. Peptides were eluted in 60% acetonitrile/0.1% TFA (Nika *et al*, 2013). Intact glycopeptides were enriched using a SAX HyperSep™ Retain AX Columns (RAX; Yang *et al*, 2017). The enrichment of intact glycopeptides using RAX facilitated efficient enzyme-substrate reaction in a small volume. Briefly, after C18 desalting, peptides in 60% acetonitrile/0.1% TFA were adjusted to 95% acetonitrile/1% TFA. The RAX column was conditioned in acetonitrile, 100 mM triethylammonium acetate, water, and finally 95% acetonitrile/1% TFA using three times per solution. Samples were loaded, washed three times using 95% acetonitrile/1% TFA, and eluted in 50% acetonitrile/0.1% TFA. The eluted intact glycopeptides were conjugated to AminoLink resin, and O-linked glycopeptides were released from the resin by incubation with OpeRATOR™ and SialEXO™ followed by C18 desalting and lyophilization as described above in the analysis of fetuin section.

### Peptide fractionation

Peptides (100 μg) were split into 96 fractions using a 1220 Series HPLC (Agilent Technologies, Inc., CA) equipped with a Zorbax Extend-C18 analytical column containing 1.8 μm particles at a flow rate of 0.3 ml/min. The mobile phase A was 10 mM ammonium formate (pH 10) and B was 10 mM ammonium formate and 90% acetonitrile (pH 10). Peptides were separated using the following linear gradient: 0–2% B, 10 min; 2–8% B, 5 min; 8–35% B, 85 min; 35–95% B, 5 min; and 95–95% B, 15 min. Fractions were collected from 0 to 96 min. The 96 fractions were concatenated into 24 fractions. The samples were then dried by lyophilization.

### LC-MS/MS analysis

Peptides dissolved in 0.1% formic acid (FA) were analyzed on a Fusion Lumos mass spectrometer with an EASY-nLC 1200 system or a Q-Exactive HF mass spectrometer (Thermo Fisher Scientific,

Bremen, Germany) with a Waters NanoAcquity UPLC (Waters, Milford, MA). The mobile phase flow rate was 0.2 μl/min with 0.1% FA/3% acetonitrile in water (A) and 0.1% FA/90% acetonitrile (B). The gradient profile was set as follows: 6% B for 1 min, 6–30% B for 84 min, 30–60% B for 9 min, 60–90% B for 1 min, and 90% B for 5 min and equilibrated in 50% B, and flow rate was 0.5 μl/min for 10 min. MS analysis was performed using a spray voltage of 1.8 kV. Spectra (AGC target $4 \times 10^5$ and maximum injection time of 50 ms) were collected from 350 to 1,800 m/z at a resolution of 60 K followed by data-dependent HCD MS/MS (at a resolution of 50 K, collision energy of 29, intensity threshold of $2 \times 10^5$, and maximum IT of 250 ms) of the 15 most abundant ions using an isolation window of 0.7 m/z. Charge-state screening was enabled to reject unassigned, single, and more than six protonated ions. Fixed first mass was 110 m/z. A dynamic exclusion time of 45 s was used to discriminate against previously selected ions. Fusion Lumos mass spectrometer was used for sequential ETD/HCD-MS2 on every precursor. In the setting of ETD-MS2, isolation mode was quadrupole with isolation window being 3. Use of calibrated charge-dependent ETD parameters was true, and detection was Orbitrap with resolution of 30 K. Maximum injection time was 250 ms, and AGC target was $10^5$. In the setting of HCD-MS2, collision energy was 36% and other parameters were the same for ETD-MS2.

### Database search of site-specific O-linked glycopeptides

Bovine fetuin (P12763) in a database with HIV gp120 (AAB50262.1) and TGFbeta1 (P01137) was used for the analysis of fetuin using the same procedure for search of the human protein database except that all fetuin peptides and four glycans including Hex(1)HexNAc(1), HexNAc, Hex(1)HexNAc(2), and Hex(2)HexNAc(2) were used. The RefSeq human protein database (72,956 sequences, downloaded from NCBI website Mar 25, 2015) was used to generate a randomized decoy database (decoy at protein level: decoy-pro) using the Trans-Proteomic Pipeline (TPP; Deutsch *et al*, 2015). The target and decoy protein database were concatenated and digested on the C-terminal side of Lys/Arg with two miss-cleavage sites (trypsin digestion) followed by digested on the N-terminal side of Ser/Thr with five miss-cleavage sites (OpeRATOR digestion) *in silico*, and finally, Ser/Thr-containing peptides with peptide lengths between 6 and 46 amino acids were used resulting in 30,759,520 non-redundant peptide entries. SEQUEST in Proteome Discoverer 2.2 (Thermo Fisher Scientific) was used to search against the database with oxidation (M), guanidination (K), Hex(1)HexNAc(1) (S/T), and HexNAc (S/T) as the variable modifications. Static modification was carbamidomethylation (C). FDR set at 1% using Percolator. MS/MS scan numbers of oxonium ion containing spectra were extracted with 10 ppm tolerance from raw files. Oxonium ion 204 was mandatory together with two of the other oxonium ions. The result was filtered to report identification with glycan modification and oxonium ions in the MS/MS spectra. Percolator generated FDR was verified using peptide identification labeled with decoy-pro in the output result.

### Label-free quantification using spectral counting

PSM numbers with 1% FDR were counted for peptides identified from tumor or normal tissue. The data were then normalized

using the total number of PSMs in individual sample. The fold change in peptides between tumor and normal samples was calculated. Peptides presented exclusively in tumor or normal tissue samples were given a two- or 0.5-fold change, respectively, in order that a fold change at the protein level could be calculated. The average fold change in peptides was calculated to obtain a fold change in proteins and at least twofold change for proteins was used. In addition, the total number of PSMs for proteins in samples was summed, and a difference in at least 10 PSM between tumor-derived and normal tissue samples was used to determine significant change.

### Bioinformatics

Site-specific O-linked glycopeptides were re-constructed into peptides of 15 amino acids in length with the O-linked glycosylation sites as the central amino acid. Online software pLogo was used to predict the motifs (O'Shea *et al*, 2013). Hierarchical clustering was used to cluster the samples based on the number of PSM of site-containing peptides using QCanvas version 1.21 (Kim *et al*, 2012). The Database for Annotation, Visualization and Integrated Discovery (DAVID) and UniProt (http://www.uniprot.org) were used for Gene Ontology (GO) analysis (Huang da *et al*, 2009).

## Data availability

The LC-MS/MS data have been deposited to the PRIDE partner repository (Vizcaino *et al*, 2016) with the dataset identifier: project accession: PXD009476 (https://www.ebi.ac.uk/pride/archive/projects/PXD009476).

**Expanded View** for this article is available online.

## Acknowledgements

We acknowledge Prof. Malcolm McCrae for proofreading, Dr. David Clark for maintenance of Fusion Lumos mass spectrometer, and Angellina Song for technical support. We acknowledge service from Johns Hopkins Mass Spectrometry and Proteomics Core. This work was supported by the National Institutes of Health, the National Institute of Allergy and Infectious Diseases (R21AI122382), National Cancer Institute, the Early Detection Research Network (EDRN, U01CA152813), the Clinical Proteomic Tumor Analysis Consortium (CPTAC, U24CA210985), National Heart Lung and Blood Institute, Programs of Excellence in Glycosciences (PEG, P01HL107153), and amfAR, The Foundation for AIDS Research on Bringing Bioengineers to Cure HIV (Grant amfAR 109551-61-RGRL). The following reagent was obtained through the NIH AIDS Reagent Program, Division of AIDS, NIAID, NIH: CEM CD4[+] cells from Dr. J.P. Jacobs.

## Author contributions

WY and HZ conceived concept and wrote the manuscript; WY conducted experimental analysis; WY, MA, and YH conducted programming, data analysis, and bioinformatics; WY and YH developed strategy for identification of glycopeptides; QKL collected and conducted pathological examination for tumor and normal kidney tissues.

## Conflict of interest

The authors declare that they have no conflict of interest.

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
