## [Review Process File · Molecular Systems Biology]

Mapping the O-Glycoproteome Using Site-specific Extraction of O-linked glycopeptides (EXoO)

Weiming Yang, Minghui Ao, Yingwei Hu, Qing Kay Li and Hui Zhang

Review timeline:

Submission date:	6 th of June 2018
Editorial Decision:	12 th of July 2018
Revision received:	22 nd of July 2018
Editorial Decision:	21 st of August 2018
Revision received:	22 nd of September 2018
Editorial Decision:	16 th of October 2018
Revision received:	16 th of October 2018
Accepted:	17 th of October 2018

Editor: Maria Polychronidou.

Transaction Report:

1st Editorial Decision

12th of July 2018

Thank you again for submitting your work to Molecular Systems Biology. We have now heard back from the two referees who agreed to evaluate your study. As you will see below, the reviewers think that the presented method seems interesting and potentially valuable for the field. They raise however a series of issues that we would ask you to address in a revision.

I think that the recommendations of the reviewers are rather clear, so there is no need to repeat the points listed below. Please contact me in case you need to further discuss any of the points raised. All issues raised by the reviewers would need to be convincingly addressed. As you might already know, our editorial policy allows in principle a single round of major revision so it is essential to provide responses to the reviewers' comments that are as complete as possible.

Reviewer #1 refers to a related paper that was published in Analytical Chemistry on June 24th 2018. According to our scoping protection policy, papers published in peer reviewed journals after the submission date of a manuscript to Molecular Systems Biology are not considered relevant to the editorial assessment of the conceptual advance/novelty of the submitted manuscript. We would ask you however to mention/discuss this related study in the revised manuscript.

Reviewer #2 mentions that the methodology is not described in enough detail. We would ask you to make sure that all information is provided and is easily accessible to the reader.

REFEREE REPORTS

Reviewer #1:

Manus Mol syst: Mapping In Vivo O-glycoproteome Using Site-specific Extraction of O-linked

glycopeptides (EXoO)

This manuscript presents an interesting new approach to O-glycoproteomics with identification of sites of O-glycans and information of the O-glycan structure attached. Glycosylation is one of the most abundant and clearly the most diverse PTMs, and for mucin-type O-glycosylation there is a clear need for new methods to define both sites and structures. The authors take advantage of a novel endo-glycopeptidase OpeRATOR, recently introduced by a company (Genovis), and combine this with immobilization of tryptic peptide digests to selectively release C-terminal peptide fragments with O-glycans attached. The OpeRATOR enzyme is not characterized in great detail, but it is proposed to cleave a peptide immediately N-terminal of an O-glycan. Sialic acids are believed to inhibit the enzyme and the authors used a sialidase (SialEXO) together with the OpeRATOR. In standard shotgun bottom-up proteomics workflow of proteolytic digests peptides dominate glycopeptides to interfere with glycopeptide identifications, and most current O-glycoproteomics approaches include an enrichment step (lectin, HILIC, hydrazide chemistry) to reduce or eliminate the peptide abundance before LC-MS/MS sequencing. The proposed approach is therefore potentially very interesting and could find wide use. The authors initially test the strategy with a single glycoprotein (fetuin) finding known glycosites (and propose a new) and then analyse complex samples including human serum, T cells and human kidney tissues. Overall the study claims to report a cumulative 3,000 O-glycosites with a third being novel. While the strategy proposed clearly warrant reporting there are a number of problems with the experimental approach and interpretations and conclusions drawn, and the manuscript text is in need of major revision. In particular, it is quite sad to see how the authors deal with the existing literature by seemingly random citations (often entirely misplaced) and omissions to drive biased interpretations. Finally, a related paper was released a few days ago and the authors now need to integrate this in the presentation (Yang S, et al. Deciphering Protein O-Glycosylation: Solid-Phase Chemoenzymatic Cleavage and Enrichment. *Anal Chem.* 2018).

Major comments:

1. The Introduction reads like a bad biased commercial and few references used to support statements are correct and appropriate. Just as an example - "The cellular machinery for O-linked glycosylation...is believed to operate stochastically in response....(ref 1). As a consequence, O-linked glycosylation can exhibit high heterogeneity". Reference 1 is a review by Ajit Varki dealing with evolutionary forces that drive diversity in glycan structures, but it does not deal with the complex biosynthetic and genetic regulation of mucin-type O-glycosylation that has perhaps the most differentially regulated initiation step to define where O-glycans are attached. The Discussion suffers similarly. The main point the authors try to drive is that their approach is the first mapping of "in vivo" O-glycoproteomes, which apart from the term being used wrongly (sialidase is used and selectivity of the OpeRATOR enzyme unknown), is an incorrect statement since the used of PNA and Jacalin lectin enrichment in a number of studies not cited (e.g. PMID: 29296958) has provided deep O-glycoproteomes with core 1 structures.
2. The authors claim that O-glycosites are unambiguously assigned by HCD. The authors first state that ETD is widely used and then that "Caveats associated with site localization using ETD render the method to be inefficient in mapping sites (refs 15,16)", which is taken out of the blue and without specifying factual issues and explanations. The data processing in the study is based on HexNAc and Hex(1)HexNAc(1) O-glycan core1 structures and the assumption that OpeRATOR cleaves immediately N-terminal to the O-glycosite. However, there are a lot of the glycopeptides identified where the Ser or Thr residues are found at positions +1 or +2 of the N-terminus, which is not discussed. Moreover, the authors do not discuss how two core1 O-glycosites or one core2 O-glycosite would affect the outcome. This reviewer suggests that the authors need to compare HCD and ETD fragmentation to make any statements. Moreover, they need to clearly state that a large number of the identified glycosites are inferred rather than unambiguously assigned with a balanced discussion of the issues at play. Approx. 95% of the presented assignments are for two simple O-glycan structures, and it is therefore a stretch to claim that the approach enables simultaneous identification of glycans and site. Adding additional glycan variables would further increase false discoveries.
3. An interesting finding is that the strategy enables identification of O-glycans in clustered regions of mucins where standard proteolytic digest strategies fail due to lack of cleavage sites. The authors should reflect on this aspect with appropriate discussion of the literature for e.g. MUC1.
4. The specificity of the OpeRATOR enzyme needs to be explored or discussion in details if the current claims of the study is to be maintained.
5. The study comparing normal and cancer tissues (pooled kidney) is premature and require knowledge of the specificity of the OpeRATOR enzyme and perhaps even considerations of the

validity in using sialidase. It is well established that many cancers involve a switch from core2 to core1 with decreased sialylation and even truncation to only GalNAc, and thus changes in glycosite identifications need to consider that the OPERATOR enzyme has preferences for different O-glycans and hence it is impossible to assign effects to sites of change of structures at individual sites.

6. The data analysis use Percolator to determine FDR. In glycoproteomics samples Percolator do not filter a lot of low score glycopeptides, where manual evaluation clearly rejects these as true cases. All Tables should include information of ppm values and Score values for the reported glycopeptides.

Reviewer #2:

Zhang and co-workers describe a novel method "Extraction of O-linked glycopeptides" EXoO and use the method to map over 3,000 O-linked glycosites to over 1,000 protein in samples from human kidney tissues, T cells and serum. The number of O-glycosites identified in total are an impressive number and in comparison to the total number of previously described O-GalNAc sites from the SimpleCell glycoproteome are more or less orthogonal. The EXoO method consists of digestion of the glycoprotein/glycoproteome, loading the peptides to a solid supported resin, application of a mucin-type O-glycopeptide protease Operator, and profiling the released glycopeptides. The method primarily combines an enrichment strategy with peptide-loaded beads with Operator, an enzyme developed by Genovis for which they have developed their own O-glycopeptide enrichment methods. There are several large advantages to using Operator, namely that the glycosite is at the first residue of the cleaved peptide thus obviating issues with glycosite localization, and it can be applied to ex vivo samples. The study here appears to be one of the larger scale applications of Operator to date applied to a set of samples of high interest. However, the authors need to clarify or expand on a number of points enumerated below. Expansion on these points will make the manuscript suitable for publication.

- There is very limited discussion of Operator and no citations to Operator literature. For example, biases in the types of glycan structure that Operator cleaves adjacent to is essential to evaluation of this method for others interested in using these enzymes.
- The use of sialidase is mentioned only in the methods and completely overlooked in the manuscript. Use of a sialidase is needed to increase efficiency of Operator cleavage, however, this means that the definition of the exact O-linked glycan structure is made ambiguous by the desialidation.
- The authors should be clear that these are "mucin-type" O-glycans throughout. Based on their data, they have primarily identified core 1 mucin type glycans, with or without sialic acids. A figure describing the types of glycan structures observed in these tissues would be very helpful to summarize this information. The authors should especially note the use of sialidase in the Discussion when asserting that "simultaneous definition of O-linked glycans at the glycosylation sites" is possible with this method.
- The methods section needs clarification: description of the enrichment from O-linked glycopeptides from tissues references trypsin digestion analogous to the previous procedure and then describes a guanidation procedure and passage over C18 followed by SAX enrichment of the glycopeptides. Operator is not mentioned in this section of the methods. This procedure is not described anywhere else and not apparent in the described procedure of EXoO at all. The need for enrichment with SAX obviates use of the solid supported resin in the first place and perhaps is a better enrichment method than the beads - some commentary should be made here.
- Did the authors do any testing of how many amino acids away from the beads the glycosite needs to be for Operator to be active?

1st Revision - authors' response

22nd of July 2018

Responses to the reviewer's comments

Reviewer #1:

Manus Mol syst: Mapping In Vivo O-glycoproteome Using Site-specific Extraction of O-linked glycopeptides (EXoO)

This manuscript presents an interesting new approach to O-glycoproteomics with identification of sites of O-glycans and information of the O-glycan structure attached. Glycosylation is one of the most abundant and clearly the most diverse PTMs, and for mucin-type O-glycosylation there is a clear need for new methods to define both sites and structures. The authors take advantage of a novel endo-glycopeptidase OpeRATOR, recently introduced by a company (Genovis), and combine this with immobilization of tryptic peptide digests to selectively release C-terminal peptide fragments with O-glycans attached. The OpeRATOR enzyme is not characterized in great detail, but it is proposed to cleave a peptide immediately N-terminal of an O-glycan. Sialic acids are believed to inhibit the enzyme and the authors used a sialidase (SialEXO) together with the OpeRATOR. In standard shotgun bottom-up proteomics workflow of proteolytic digests peptides dominate glycopeptides to interfere with glycopeptide identifications, and most current O-glycoproteomics approaches include an enrichment step (lectin, HILIC, hydrazide chemistry) to reduce or eliminate the peptide abundance before LC-MS/MS sequencing. The proposed approach is therefore potentially very interesting and could find wide use. The authors initially test the strategy with a single glycoprotein (fetuin) finding known glycosites (and propose a new) and then analyse complex samples including human serum, T cells and human kidney tissues. Overall the study claims to report a cumulative 3,000 O-glycosites with a third being novel. While the strategy proposed clearly warrant reporting there are a number of problems with the experimental approach and interpretations and conclusions drawn, and the manuscript text is in need of major revision. In particular, it is quite sad to see how the authors deal with the existing literature by seemingly random citations (often entirely misplaced) and omissions to drive biased interpretations. Finally, a related paper was released a few days ago and the authors now need to integrate this in the presentation (Yang S, et al. Deciphering Protein O-Glycosylation: Solid-Phase Chemoenzymatic Cleavage and Enrichment. Anal Chem. 2018).

Response: We appreciate the reviewer's suggestions and revised the introduction, experimental approach, interpretations, conclusions, and citations of the manuscript. Please see the specific responses to each of the reviewer's specific comments below.

Thanks for reviewer's suggestion on the recent publication of Yang S, et al. During the peer-review period, we noticed the publication in Analytical Chemistry (AC). We have informed the editor "that a June 24th publication by Analytical Chemistry used the approach described in our submitted article." Since we submitted our manuscript to MSB on June 6th, our submission of manuscript was prior to the publication by Analytical Chemistry. We checked publication policy by Molecular Systems Biology for Scooping protection and found that "A manuscript submitted to Molecular Systems Biology is subject to scooping protection from the day of submission to Molecular Systems Biology and extends through the agreed revision period." And "Please inform editors as soon as you become aware of other studies directly relevant to your study. Conceptually related studies formally published elsewhere must be cited (citations can be added at proofs stage where necessary)." The publication described by Yang et al is relevant to our study but in a quick and sample system, at the similar level as the proof of concept study described in Figure 1 of our submitted article. Our manuscript went further to use the developed O-glycoproteomic approach to analyze cells, plasma, and tissues and identified over 3,000 O-linked glycosites, the largest number of glycosites identified in any published studies. During the revision of this manuscript, we have included this publication (page14, line 309).

Major comments:

1. The Introduction reads like a bad biased commercial and few references used to support statements are correct and appropriate. Just as an example - "The cellular machinery for O-linked glycosylation...is belived to operate stochastically in response....(ref 1). As a consequence, O-linked glycosylation can exhibit high heterogeneity". Reference 1 is a review by Ajit Varki dealing with evolutionary forces that drive diversity in glycan structures, but it does not deal with the complex biosynthetic and genetic regulation of mucin-type O-glycosylation that has perhaps the most differentially regulated initiation step to define where O-glycans are attached. The Discussion suffers similarly. The main point the authors try to drive is that their approach is the first mapping of "in vivo" O-glycoproteomes, which apart from the term being used wrongly (sialidase is used and selectivity of the OpeRATOR enzyme unknown), is an incorrect statement since the used of PNA and Jacalin lectin enrichment in a number of studies not cited (e.g. PMID: 29296958) has provided deep O-glycoproteomes with core 1 structures.

Response: as suggested by reviewer, we have revised the introduction and discussion to include additional citations. In addition, "The presence of up to 20 GalNAc-transferases (GalNAc-Ts) for adding the initial sugar to amino acid residues in different sequence regions further complicates the

dynamic regulation of O-linked glycosylation” was included to further describe the heterogeneity regulated by the initiation step to define where O-glycans are attached (page 3, line 57-59). As suggested, we have revised the manuscript to cite PMID: 29296958 in the introduction (Page 4, line 66).

Reviewer has concerns of “in vivo”. We agreed with reviewer and deleted “in vivo”. The study does not claim to be the first to map in vivo O-linked glycoproteome.

The reviewer has concern on the use of sialidase. We have revised the manuscript to include discussion “The use of sialidase in the procedure facilitated efficient cleavage by OpeRATOR that was used to improve mapping of O-linked glycosylation sites. The addition of sialidase could be omitted if the study focus is to define site-specific glycans with sialic acid” (Page 13 line 269-271 in revised manuscript).

2. *The authors claim that O-glycosites are unambiguously assigned by HCD. The authors first state that ETD is widely used and then that "Caveats associated with site localization using ETD render the method to be inefficient in mapping sites (refs 15,16)", which is taken out of the blue and without specifying factual issues and explanations. The data processing in the study is based on HexNAc and Hex(1)HexNAc(1) O-glycan core1 structures and the assumption that OpeRATOR cleaves immediately N-terminal to the O-glycosite. However, there are a lot of the glycopeptides identified where the Ser or Thr residues are found at positions +1 or +2 of the N-terminus, which is not discussed. Moreover, the authors do not discuss how two core1 O-glycosites or one core2 O-glycosite would affect the outcome. This reviewer suggests that the authors need to compare HCD and ETD fragmentation to make any statements. Moreover, they need to clearly state that a large number of the identified glycosites are inferred rather than unambiguously assigned with a balanced discussion of the issues at play. Approx. 95% of the presented assignments are for two simple O-glycan structures, and it is therefore a stretch to claim that the approach enables simultaneous identification of glycans and site. Adding additional glycan variables would further increase false discoveries.*

Response: As suggested by the reviewer, we have revised the manuscript to describe factual issues associated with ETD in the introduction “Mass spectrometry (MS) using electron transfer dissociation (ETD) for fragmentation has to-date been the prevailing analytical tool to localize O-linked glycosylation sites (Zhu et al, 2013). However, the site localization using ETD requires precursor ions with high charge states, and presence of convincing peptide fragment ions covering the O-linked glycosylation sites, rendering the method to be inefficient in mapping certain O-linked glycosylation sites (Darula et al, 2012; Good et al, 2007; Mulagapati et al, 2017)” (Page 4, line 73-78). The explanations have been detailed in other review papers that may not need further explanation in this manuscript.

Reviewer has concern for the Ser and Thr residues found at position +1 and +2 of the N-terminus that may confound the ambiguity of localization of the O-linked glycosylation sites. In the proof of concept study of fetuin, we saw that OpeRATOR cleaved immediately N-terminal to the known O-glycosite but not other Ser/Thr without O-glycan. The cleavage specificity is also documented in the company’s experimental data studying other proteins.

We agreed with reviewer that two core 1 O-glycosites or one core 2 O-glycosites might confound the data for definition of site-specific glycans. We further explored our data and found that approximately 69% of total PSM contained a single core 1 glycan composition to support the definition of the glycan on those site-containing glycopeptides. For other glycan compositions that could be the result of multiple glycans on a glycopeptide, caution must be taken in the data interpretation. We have revised the manuscript accordingly to discuss these concerns in the discussion section “In addition, possibility of multiple glycans on a glycopeptide demands caution in the data interpretation to define site-specific glycan composition. For example, two Hex(1)HexNAc(1) on a glycopeptide could yield a glycan composition of Hex(2)HexNAc(2) in the result.” (Page 13, 284-287).

We agreed with reviewer that adding additional glycan variables would further increase false discoveries.

3. *An interesting finding is that the strategy enables identification of O-glycans in clustered regions of mucins where standard proteolytic digest strategies fail due to lack of cleavage sites. The authors should reflect on this aspect with appropriate discussion of the literature for e.g. MUC1.*

Response: as suggested, we have revised the manuscript to discuss the importance of MUC1 in different diseases in the discussion section “Therefore, using EXoO allowed detailed mapping of over one hundred sites on VCAN and MUC1. MUC1 has been reported to be important molecule in many research areas including different cancers, immunity and immunotherapy (Beatson et al, 2016;

Hanisch, 2005; Hanson & Hollingsworth, 2016). The use of EXoO therefore is advantageous to reveal new biological insight regarding mucin-type glycoproteins” (Page 14, line 298-302).

4. The specificity of the OpeRATOR enzyme needs to be explored or discussion in details if the current claims of the study is to be maintained.

Response: We agree with the reviewer’s concern on the specificity of OpeRATOR and included discussion in the revised manuscript “The precise specificity of OpeRATOR for different O-linked glycans remains unclear. Analysis of our data from tissue, serum and cells revealed that approximately 69% of total PSM contained glycan composition Hex(1)HexNAc(1) that was most likely to be core 1 mucin type glycan GalGalNAc. Therefore, it is possible to define that the O-linked glycopeptide contained Hex(1)HexNAc(1) or most likely to be GalGalNAc with or without sialic acid at the site of O-linked glycosylation. This data could also be explained as that the major glycan composition for site-specific O-linked glycopeptide is the core 1 structure Hex(1)HexNAc(1) that is prevalent in a wide range of glycoproteins from different cell types compared to the relatively restricted presence of other core structures seen in specific tissue and cell types (Brockhausen & Stanley, 2015). However, the fact that other glycoforms accounted for approximately 31% of total identified glycan compositions argues that further investigation is needed to definitively establish the glycoform specificity of OpeRATOR” (page 13, line 273-284). In addition, the specificity of OpeRATOR for Ser/Thr with O-glycans was characterized by the company and showed in our proof of principle study of fetuin.

5. The study comparing normal and cancer tissues (pooled kidney) is premature and require knowledge of the specificity of the OpeRATOR enzyme and perhaps even considerations of the validity in using sialidase. It is well established that many cancers involve a switch from core2 to core1 with decreased sialylation and even truncation to only GalNAc, and thus changes in glycosite identifications need to consider that the OpeRATOR enzyme has preferences for different O-glycans and hence it is impossible to assign effects to sites of change of structures at individual sites.

Response: We agreed with the reviewer that change of glycoform is associated with the cancer tissues. Prior to this study, limited information is known toward the O-linked glycoproteome and the sites in tissue due to a lack of suitable technology. This study mapped the site-specific of O-linked glycoproteome and pave the path to further investigation that reveals aberrant O-linked glycans at specific site in cancer tissues. In a direct comparison between cancer and normal tissues with the same workflow, the study provided new insight into site-specific O-linked glycoproteome in kidney cancer. We agreed that specificity of OpeRATOR is an important consideration and have revised the manuscript to discuss the specificity of OpeRATOR in the discussion section. The effects associated with specific glycoprotein in discussion are reported by literature, we have lower down the tone in the manuscript to described that the change of sites may be associated with these effects.

6. The data analysis use Percolator to determine FDR. In glycoproteomics samples Percolator do not filter a lot of low score glycopeptides, where manual evaluation clearly rejects these as true cases. All Tables should include information of ppm values and Score values for the reported glycopeptides.

Response: As requested by reviewer, we have revised the data to include the ppm values i.e. DeltaM [ppm] and score value i.e. Percolator SVMscore reported in SEQUEST from Proteome Discoverer 2.2 for the identified glycopeptides. Other scores and Percolator values are also included. We have manually inspected few hundreds of identified glycopeptides with different scores using spectral viewer in the Proteome Discoverer 2.2 and were sure about the identification with the current search settings.

Reviewer #2:

Zhang and co-workers describe a novel method "Extraction of O-linked glycopeptides" EXoO and use the method to map over 3,000 O-linked glycosites to over 1,000 protein in samples from human kidney tissues, T cells and serum. The number of O-glycosites identified in total are an impressive number and in comparison to the total number of previously described O-GalNAc sites from the SimpleCell glycoproteome are more or less orthogonal. The ExoO method consists of digestion of the glycoprotein/glycoproteome, loading the peptides to a solid supported resin, application of a mucin-type O-glycopeptide protease Operator, and profiling the released glycopeptides. The method primarily combines an enrichment strategy with peptide-loaded beads with Operator, an enzyme developed by Genovis for which they have developed their own O-glycopeptide enrichment methods. There are several large advantages to using Operator, namely that the glycosite is at the first residue of the cleaved peptide thus obviating issues with glycosite localization, and it can be applied

to ex vivo samples. The study here appears to be one of the larger scale applications of Operator to date applied to a set of samples of high interest. However, the authors need to clarify or expand on a number of points enumerated below. Expansion on these points will make the manuscript suitable for publication.

- There is very limited discussion of Operator and no citations to Operator literature. For example, biases in the types of glycan structure that Operator cleaves adjacent to is essential to evaluation of this method for others interested in using these enzymes.

Response: For discussion of glycan specificity of OpeRATOR, as suggested by this reviewer and previous reviewer, we have explored our data and revised the manuscript to include a section to discuss this issue. “The precise specificity of OpeRATOR for different O-linked glycans remains unclear. Analysis of our data from tissue, serum and cells revealed that approximately 69% of total PSM contained glycan composition Hex(1)HexNAc(1) that was most likely to be core 1 mucin type glycan GalGalNAc. Therefore, it is possible to define that the O-linked glycopeptide contained Hex(1)HexNAc(1) or most likely to be GalGalNAc with or without sialic acid at the site of O-linked glycosylation. This data could also be explained as that the major glycan composition for site-specific O-linked glycopeptide is the core 1 structure Hex(1)HexNAc(1) that is prevalent in a wide range of glycoproteins from different cell types compared to the relatively restricted presence of other core structures seen in specific tissue and cell types (Brockhausen & Stanley, 2015). However, the fact that other glycoforms accounted for approximately 31% of total identified glycan compositions argues that further investigation is needed to definitively establish the glycoform specificity of OpeRATOR” (page 13, line 273-284).

In introduction session, we also introduced OpeRATOR as “OpeRATOR, identified from the mucin degrading human intestinal bacterium *Akkermansia muciniphila*, recognizes O-linked glycans and cleaves O-linked glycopeptides at the N-termini of O-linked glycan-occupied Ser or Thr to release site-specific O-linked glycopeptides with the glycosylation sites at the N-terminus of the peptide for unambiguous localization” (Page 5, line 98-102).

In the proof of concept study of fetuin with known O-linked glycosylation sites, the cleavage specificity of OpeRATOR has been demonstrated. In addition, the company has established the cleavage specificity by studying other glycoproteins.

- The use of sialidase is mentioned only in the methods and completely overlooked in the manuscript. Use of a sialidase is needed to increase efficiency of Operator cleavage, however, this means that the definition of the exact O-linked glycan structure is made ambiguous by the desialidation.

Response: We agreed with the reviewer and revised the manuscript to discuss this issue. In the discussion, we added “The use of sialidase in the procedure facilitated efficient cleavage by OpeRATOR that was used to improve mapping of O-linked glycosylation sites. The addition of sialidase could be omitted if the study focus is to define site-specific glycans with sialic acid” (Page 13 line 269-271 in revised manuscript).

- The authors should be clear that these are "mucin-type" O-glycans throughout. Based on their data, they have primarily identified core 1 mucin type glycans, with or without sialic acids. A figure describing the types of glycan structures observed in these tissues would be very helpful to summarize this information. The authors should especially note the use of sialidase in the Discussion when asserting that "simultaneous definition of O-linked glycans at the glycosylation sites" is possible with this method.

Response: We agreed with the reviewer that the identified O-glycans were primarily core 1 mucin type glycans. We have revised the manuscript to discuss “The precise specificity of OpeRATOR for different O-linked glycans remains unclear. Analysis of our data from tissue, serum and cells revealed that approximately 69% of total PSM contained glycan composition Hex(1)HexNAc(1) that was most likely to be core 1 mucin type glycan GalGalNAc.” Other glycan compositions were detected. However, HCD-MS is not able to identify the precise glycan structures. Instead, glycan compositions were provided in the supplement tables. In the case that there is a single glycan composition being attached to peptide such as Hex(1)HexNAc(1), it is possible to define that the O-linked glycopeptide contained Hex(1)HexNAc(1) or most likely to be GalGalNAc with or without sialic acid at the site of O-linked glycosylation. We have revised the manuscript to include “Therefore, it is possible to define that the O-linked glycopeptide contained Hex(1)HexNAc(1) or most likely to be GalGalNAc with or without sialic acid at the site of O-linked glycosylation.” in the discussion (Page 13, line 273-278).

- *The methods section needs clarification: description of the enrichment from O-linked glycopeptides from tissues references trypsin digestion analogous to the previous procedure and then describes a guanidation procedure and passage over C18 followed by SAX enrichment of the glycopeptides. Operator is not mentioned in this section of the methods. This procedure is not described anywhere else and not apparent in the described procedure of ExoO at all. The need for enrichment with SAX obviates use of the solid supported resin in the first place and perhaps is a better enrichment method than the beads - some commentary should be made here.*

Response: as suggested, we have revised this method section to clarify that proteins from human kidney tissues, serum and CEM T cells were trypsin-digested as described in the analysis of fetuin section. The guanidination and desalting of peptides were conducted on a C18 column (Nika et al, 2013). Intact glycopeptides were enriched using RAX column (Yang et al, 2017). In addition, we have revised the manuscript to include the use of solid-phase and Operator after enrichment of intact glycopeptides by SAX. Furthermore, we have revised the manuscript to explain the use of SAX. "The enrichment of intact glycopeptides using RAX facilitated efficient enzyme-substrate reaction in a small volume" (Page 17, line 361-363). SAX also enriches other hydrophilic peptides and non-glycopeptide contaminants. We believe that the use of solid-phase and Operator will be efficient to remove contaminants.

- *Did the authors do any testing of how many amino acids away from the beads the glycosite needs to be for Operator to be active?*

Response: We have not tested how many amino acids away from the beads the glycosite needs to be for Operator to be active.

Editorial revision:

Reviewer #1 refers to a related paper that was published in Analytical Chemistry on June 24th 2018. According to our scooping protection policy, papers published in peer reviewed journals after the submission date of a manuscript to Molecular Systems Biology are not considered relevant to the editorial assessment of the conceptual advance/novelty of the submitted manuscript. We would ask you however to mention/discuss this related study in the revised manuscript.

Response: Please see the response to reviewer #1.

Reviewer #2 mentions that the methodology is not described in enough detail. We would ask you to make sure that all information is provided and is easily accessible to the reader. Related to this point, we are piloting "Structured Methods" a new format for the Materials and Methods. Adhering to this format is mandatory for the Method article type and for papers with a strong methodological focus. Specifically, the Material and Methods section should include a Reagents and Tools Table (listing key reagents, experimental models, software and relevant equipment and including their sources and relevant identifiers) followed by a Methods and Protocols section in which we encourage the authors to describe their methods using a step-by-step protocol format with bullet points. More information on how to adhere to this format as well as downloadable templates (.doc or .xls) for the Reagents and Tools Table can be found in our author guidelines:

<http://msb.embopress.org/authorguide#researcharticleguide>. An example of a Method paper with Structured Methods can be found here: <http://msb.embopress.org/content/14/7/e8071>.

Response: We have included a Reagents and Tools Table with list of key reagents.

2nd Editorial Decision

21st of August 2018

Thank you again for sending us your revised manuscript. We have now heard back from reviewer #1 who was asked to evaluate your study. As you will see below, this reviewer still raises some remaining issues, which we would ask you to address in a revision.

Most of the remaining issues can be addressed by text modifications (e.g. mentioning the Yang et al, 2018 Anal Chem study in the introduction, correcting the numbers of mapped glycosylation sites, correcting several citations etc.) and other relatively minor revisions. However, one somewhat more major point refers to the need to perform a direct comparison of ETD vs. HCD, which we would ask you to include in the revision.

 REFEREE REPORTS

Reviewer #1:

The revised version of the manuscript does not satisfactorily meet the criticism raised by this reviewer and for that sake the common concerns of both reviewers, even though the rebuttal letter seems to suggest so. The entire manuscript and presentation/interpretation of data has to be reorganized/reformulated and placed in appropriate and balanced light of the current literature including the recent report in *Anal Chem*. The authors would need to revisit the previous comments raised by both reviewers, but below are some specific points of concern:

The numbers game with 3,000 sites and 1,000 proteins still used has to be revised given the ambiguities pointed out with the MS identifications. The statement p7 of unambiguously mapped sites does not seem to be correct?

The Introduction is still highly biased and trying to lead up to a new method without reduction in sample complexity and with information of natural glycan structures. Since this is the holy grail of the field it is important upfront to bring the reader into the reality that the methods at best can report only core1 structures after removal of sialic acids. The authors still fail to discuss that current lectin enrichments and ETD MS of desialylated core1 glycoproteomes reach >1-2,000 O-glycosites in plasma and organs? See e.g. PMID: 29296958?

Furthermore, the Introduction should discuss the *Anal Chem* paper with specific information (not currently without specifics at the end of the Discussion) and fully disclose available information of the substrate specificity of the Operator enzyme.

The statement "...sialidase could be omitted if the focus is to define site specific glycans with sialic acids", needs further qualification? The producer recommends use of sialidase presumably due to efficiency reasons, but without data this could at best only be a hypothesis that needs to be tested? As commented previously the discussion of ETD vs HCD needs to be balanced and include pitfalls with HCD. Without direct comparison (as suggested) the conclusions drawn may not be true. This reviewer's experience is that current MS with OrbiTraps provide similar identification numbers for glycopeptides in both ETD and HCD modes, although the datasets may vary. It is correct that for ETD MS/MS precursor ions has to be multiply charged and the fragmentation pattern is better when the charge is more than 3. However, precursor ions with $z=2+$ still provide decent fragmentation patterns and in our experience the majority of glycopeptides have charge states of 3 or 4 in any case. Thus, the key issue of unambiguous assignments with HCD needs to be discussed. The authors still claim that they have unambiguous site assignments.

The concern about Ser/Thr found at positions +1 and +2 was not addressed appropriately. The simple fetuin model may not reflect the performance with the diversity of the entire proteome. Here, running the same sample in both HCD and ETD would give an answer.

If the Operator specificity is unknown it may be premature to use it to explore aberrant glycoproteomes (p11) without discussion of the limitations - namely that increase in core1 O-glycans with sialic acid capping is likely the only property measured and interpretations of how this comes about may be difficult?

The final conclusion that "EXoO identified a substantially larger number of sites and glycoproteins almost doubling of the number of sites identified in decades" is incorrect and the authors need to survey the literature for recent reports using different lectin enrichments with cell lines, plasma and even organs.

Finally, many citations are still random and in places completely out of scope.

6) Authors agree that only 69% of their data has glycan composition to be clear associated with one structure while the rest 31% cases glyco site numbers could be over estimated. On the other hand the total number of glycosites still the same (3000).

The authors responded that manual validation was performed and false cases were filtered out. It is difficult to judge (very subjective and based on the user experience), but just as an example in Dataset EV1 cases with the Xcore value below 1 were found. It is especially strange because some of these cases were reported heavily glycosylated. For example peptide SAVPDAA....VGP with 6 monosaccharides per sequence with the Xcore 0,8. This is very hard to accept as a true identification. This reviewer would recommend to include a supplemental annotated spectra. Proteome Discoverer has this function.

Response to reviewer's comments:*Reviewer#1:*

The revised version of the manuscript does not satisfactorily meet the criticism raised by this reviewer and for that sake the common concerns of both reviewers, even though the rebuttal letter seems to suggest so. The entire manuscript and presentation/interpretation of data has to be reorganized/reformulated and placed in appropriate and balanced light of the current literature including the recent report in Anal Chem. The authors would need to revisit the previous comments raised by both reviewers, but below are some specific points of concern:

1. The numbers game with 3,000 sites and 1,000 proteins still used has to be revised given the ambiguities pointed out with the MS identifications. The statement p7 of unambiguously mapped sites does not seem to be correct?

Reponses: In the response below, we have provided experimental data and explanation to address each of the sources of the ambiguities (provided below in responses to the specific questions on this topic). Given the ambiguities with the MS identifications, we revised the manuscript and deleted the “unambiguously” in page 7 and other places in the manuscript.

2. The Introduction is still highly biased and trying to lead up to a new method without reduction in sample complexity and with information of natural glycan structures. Since this is the holy grail of the field it is important upfront to bring the reader into the reality that the methods at best can report only core1 structures after removal of sialic acids. The authors still fail to discuss that current lectin enrichments and ETD MS of desialylated core1 glycoproteomes reach >1-2,000 O-glycosites in plasma and organs? See e.g. PMID: 29296958?

Response: we apologize for misunderstanding the reviewer may have. We did not try to lead up a new method without reduction in sample complexity. In fact, we needed to reduce sample complexity by using SAX column to enrich intact O-linked glycopeptides. Next, the use of solid-phase support was needed to further capture peptides and reduce sample complexity via specific release of site-specific O-linked glycopeptides. We revised the introduction by replacing the sentence “All of these methodologies have reduced the sample complexity by enrichment of O-linked glycopeptides from background peptides, which remains as a severe hindrance to the structural and functional study of O-linked glycoproteins.” with the sentence “The enrichment methodologies have proved to be useful to study O-linked glycoproteome in different biological systems.” In p4 line 69-70.

Reviewer has concern regarding introduction that our method can report all natural glycan structures instead of reporting core 1 structures after removal of sialic acids. To address this concern, we revised the introduction by replacing the sentence “It has been designed to simultaneously enrich and identify O-linked glycosylation sites and define their site-specific glycans” with the sentence to “It has been designed to simultaneously enrich and identify O-linked glycosylation sites and define their site-specific glycans with primarily core 1 structures with or without sialic acids” in p5 line 98-100. The EXoO method can work on core 1 structures without removal of sialic

acids. Please refer to response to reviewer's comment #3 below for the data and discussion regarding the use of sialidase.

Reviewer requests to further discuss PMID: 29296958 regarding lectin enrichment and ETD-MS analysis of glycoproteome in plasma and organs in PMID: 29296958. In our previous manuscript, we have cited this article with several others on the O-glycoproteomic methods using lectins. As requested by the reviewer, in this revision, we described the article, PMID: 29296958, in details. We have re-checked: Characterizing the O-glycosylation landscape of human plasma, platelets, and endothelial cells. In the abstract of the paper, "native tissue" has been mentioned. However, the paper only studied plasma, platelets and endothelial cells mentioned in the title and described in the Materials and Methods section: "AB RhD-positive platelets from 4 random donors and plasma were obtained from the Blood Bank of the Capital Region and harvested according to standard protocols. Primary Human Umbilical Vein Endothelial Cells (HUVEC) were purchased from Life Technologies." The HUVEC is cells cultured in medium 200PRF supplemented with low serum growth supplement as described in the Supplement Methods of PMID: 29296958. Therefore, there is no data for organ or tissue in the study described in PMID: 29296958.

Regarding the plasma data described in article PMID: 29296958, the authors described "the hitherto largest O-glycoproteome", among the total number of 1,123 O-glycosites reported in this study, 354 O-glycosites were identified from plasma. We therefore revised the manuscript introduction in p4 line 77-80 to describe the study: "In the largest O-glycoproteome study, lectin enrichments of de-sialylated core 1 O-linked glycopeptides from plasma using PNA and VVA and ETD-MS2 analysis, 354 unique O-linked glycosylation sites were reported among the total of 1,123 O-glycosites identified from plasma, platelets, and endothelial cells (King et al, 2017)."

3. Furthermore, the Introduction should discuss the Anal Chem paper with specific information (not currently without specifics at the end of the Discussion) and fully disclose available information of the substrate specificity of the Operator enzyme.

Response: we agree with reviewer. Therefore, we have surveyed the data produced from the company and also in the Yang et al. Anal Chem 2018.

Specifically, information from the company shows that OpeRATOR is active on sialylated O-glycoproteins (lane 2 in figure below). In lane 2, TNF α receptor can be digested by OpeRATOR without sialidase treatment. Lane 3 shows that the TNF α receptor contain sialic acids and treatment of sialidase removed sialic acids from the protein and decreased the molecular weight.

Effect of SialEXO Pretreatment on OpeRATOR Enzymatic Activity

Figure 1. OpeRATOR is active on sialylated O-glycoproteins, but the activity is higher when the sialic acids are removed using SialEXO. The native O-glycosylated TNF α receptor (TNF α R) was incubated with OpeRATOR with or without the addition of SialEXO overnight, and the reactions were separated on SDS-PAGE.

In addition, information from poster from the company shows that protein can be digested with OpeRATOR without sialidase (red square in the workflow below) that leads to identification of a glycopeptide S.TSFLPMGP.S with HexHexNAcNeuAc (red

square in the table below). The figure below is cut and recognized from company's poster.

Figure 2. Workflow for site-specific determination of O-glycan sites. OgpA hydrolyzes the substrate N-terminally of O-glycosylated sites. Addition of Sialidase mix (SialEXO™, Genovis) is optional but improves digestion. PNGaseF (Sigma) is added to reduce heterogeneity, and as an option in a second step, HexNAcHex can be removed from the asialylated glycopeptides using an O-glycosidase from *Streptococcus oralis* (OglyZOR™, Genovis).

Range	Sequence	G. Composition	Mass Measured	Mass Theoretical	Δppm
184 - 198	PTRSMAPGAVHLPOPVS	-	1559.813	1559.824	-7.12
186 - 199	RSMAPGAVHLPOPVS.T	-	1405.699	1405.702	-2.13
186 - 204	RSMAPGAVHLPOPVSTRSQH.T	2 x -[O]	2729.256	2729.270	-5.06
200 - 207	S.TRSQHTQPT	-	953.470	953.468	+1.99
205 - 207	H.TQPT	-	344.172	344.170	+5.81
208 - 216	PTPEPSTAPS.T	3 x -[O]	1980.796	1980.805	+4.24
212 - 216	P.STAPS.T	-	461.213	461.212	+1.95
217 - 225	S.TSFLPMGFS	-[O]-[O]	1618.701	1617.693	+17.62

Original section from the poster:

Applications of the O-protease in LC-MS

The unique specificity for O-glycosylated residues on glycoproteins opens for a variety of applications using this new OgpA in the glycoproteomic field. By using the workflow illustrated in Fig. 2, the glycosylation sites in the highly O-glycosylated region of the biopharmaceutical etanercept, were assessed. A summary of the obtained data is presented in Fig. 3. This work is performed in solution, but recently, new techniques have been described, where tryptic peptides are conjugated to solid support prior digestion of OgpA (Ref. 1).

Figure 3. OgpA digestion of heavily O-glycosylated region of etanercept. a) The specificity of OgpA enabled site-specific determination of O-glycosylation. b) Selected mass data of O-glycosylated peptides acquired from RP-C18 and HILIC separation prior ESI-Q-TOF-MS analysis.

Figure 2. Workflow for site-specific determination of O-glycan sites. OgpA hydrolyzes the substrate N-terminally of O-glycosylated sites. Addition of Sialidase mix (SialEXO™, Genovis) is optional but improves digestion. PNGaseF (Sigma) is added to reduce heterogeneity, and as an option in a second step, HexNAcHex can be removed from the asialylated glycopeptides using an O-glycosidase from *Streptococcus oralis* (OglyZOR™, Genovis).

Moreover, in data from Yang et al. Anal Chem 2018 where bovine submaxillary gland mucin (MBS) O-glycosites are studied. As the authors stated that “We compared the identification of O-glycosites with and without sialic acids by treating with neuraminidase (Supporting Information Table S3).” We checked their supporting information table S3 and filtered to display glycopeptides with sialic acid (table below). Their data appeared to show glycans with NeuAc and/or NeuGc. For example, glycopeptide K.tSQQL.S with Hex(1)HexNAc(1)NeuAc(1). We also see L.sVR.V with Hex(2)HexNAc(2)NeuAc(2). Given no data analysis section in their paper, we are not confident to comment that the OperATOR can recognize core 2 structures because sVR contains only one O-glycosite in the sequence and the glycan composition can only be interpreted to be a core 2 structure.

Supplementary Table S3. List of unique O-glycopeptides identified by O-GIG on mucin bovine submaxillary gland without removal of sialic acids. The missed-cleavage is set as "1". Fifty-seven unique O-glycopeptides are identified from MSB without removal of sialic acids, consisting of 37 O-glycosites.

O-glycopeptide	O-glycan	O-glycosite	Area	Abundance
S.sR.A	HexNAc(1)Hex(1)NeuAc(1)	313	2.02E+03	0.0000%
K.tSQQL.S	HexNAc(1)Hex(1)NeuAc(1)	361	2.31E+07	0.1148%
K.TsQQL.S	HexNAc(1)Hex(1)NeuAc(1)	362	5.78E+06	0.0287%
R.ItNLQFN.S	HexNAc(1)Hex(1)NeuAc(1)	379	9.30E+06	0.0462%
L.sEIK.F	HexNAc(1)Hex(1)NeuAc(1)	419	1.54E+06	0.0076%
I.tNLQFN.S	HexNAc(1)Hex(1)NeuAc(1)NeuGc	379	6.76E+07	0.3355%
V.tSAHK.G	HexNAc(1)Hex(1)NeuAc(2)	305	4.24E+06	0.0210%
V.TsAHK.G	HexNAc(1)Hex(1)NeuAc(2)	306	1.70E+07	0.0842%
V.sK.G	HexNAc(1)Hex(1)NeuAc(2)	321	2.53E+06	0.0126%
R.tAAS.S	HexNAc(1)Hex(1)NeuGc(1)	344	4.52E+06	0.0224%
S.sPSIAL.S	HexNAc(1)Hex(1)NeuGc(1)	348	7.29E+05	0.0036%
L.sVRV.S	HexNAc(1)Hex(1)NeuGc(1)	366	6.59E+06	0.0327%
L.sEIKFR.P	HexNAc(1)Hex(1)NeuGc(1)	419	3.63E+05	0.0018%
G.sVVVEL.T	HexNAc(1)Hex(1)NeuGc(1)	427	2.58E+07	0.1281%
L.sVR.V	HexNAc(2)Hex(2)NeuAc(2)	366	7.69E+06	0.0382%
H.tHGR.Y	HexNAc(2)Hex(2)NeuAc(2)	541	1.25E+06	0.0062%

In the previous study published by AC, the cleavage specificity of OperATOR was not examined. As requested by this reviewer, we revised the manuscript to discuss the Anal Chem paper in the introduction in p5 line 107-112: "During the peer-review period of our study, a manuscript by Yang et al, (Yang et al, 2018), described the analysis of O-linked glycosylation sites from several simple glycoproteins including fetuin, mucin, and Zika viral proteins. This study took use of the enzyme, Operator, with cleavage of peptide sequences at the N-termini of the O-linked glycosylation sites (Yang et al, 2018). However, the glycan specificity and cleavage specificity for the O-linked glycosylation sites by Operator enzyme is not clearly defined."

The reviewer asks to fully disclose available information of the substrate specificity of the Operator enzyme. With our study described in the original manuscript using HCD-MS2 and the additional ETD-MS2 data described in the revised manuscript, we have disclosed the enzymatic specificity information supported by data presented in this study and to the best of our knowledge.

4. The statement "sialidase could be omitted if the focus is to define site specific glycans with sialic acids", needs further qualification? The producer recommends use of sialidase presumably due to efficiency reasons, but without data this could at best only be a hypothesis that needs to be tested?

Response: The above experimental data in response to the reviewer's comment #3 support that the O-linked glycosylation sites containing sialic acids O-glycans can be released by OperATOR. However, the company also recommends removing the sialic acids to increase the efficiency of cleavage by OperATOR.

Therefore, we have revised the manuscript and credited the original recommendation by the company. In p15 324-327: "As stated by manufacture, the use of sialidase in the procedure facilitated efficient cleavage by OperATOR that was used to improve mapping of O-linked glycosylation sites. The addition of sialidase could be omitted if the study focus is to define site-specific glycan structures with sialic acid."

5. As commented previously the discussion of ETD vs HCD needs to be balanced and include pitfalls with HCD. Without direct comparison (as suggested) the conclusions drawn may not be true. This reviewer's experience is that current MS with Orbitraps provide similar identification numbers for glycopeptides in both ETD and HCD modes, although the datasets may vary. It is correct that for ETD MS/MS precursor ions has to be multiply charged and the fragmentation pattern is better when the charge is more than 3. However, precursor ions with $z=2+$ still provide decent fragmentation patterns and in our experience the majority of glycopeptides have charge states of 3 or 4 in any case. Thus, the key issue of unambiguous assignments with HCD needs to be discussed.

Response: we apologize for the confusion, but we did not compare and discuss ETD vs HCD in the previous revision of the manuscript.

In study of tryptic glycopeptides, while ETD and HCD may identify similar number for glycopeptides. The site localization will have to rely on ETD data. HCD cannot confidently assign O-glycosites in the glycopeptides due to preferential fragmentation of glycans in the HCD mode. To reflect this fact, we revised the manuscript in the introduction to state in p4 80-83: "Alternative to ETD fragmentation, HCD-MS2 may provide an efficient fragmentation method to identify glycopeptides but the O-linked glycosylation sites cannot be confidently assigned due to preferential fragmentation of O-linked glycans during HCD mode (Yang et al, 2014)."

Prior to this study, there is no ETD data on the type of glycopeptides cleaved by OperATOR. Site-specific O-glycopeptides generated by trypsin and OperATOR can be shorter than tryptic glycopeptides that many of the precursors have +2 charge. As an experimental evidence, we examined our data and found that 38% PSMs in the tissue data are glycopeptides with +2 charges. The precursor with +2 charges may not be efficiently fragmented in the ETD mode. To have a discussion of ETD vs HCD on glycopeptides generated by OperATOR as requested by the reviewer, we conducted additional experiment to use sequential ETD/HCD and wrote an addition result section in p8 line 162-201: **Specificities of OperATOR for peptides and O-linked glycans**. In this additional result section and the addition dataset EV5, we found that HCD-MS2 identified 85 unique glycopeptides and ETD-MSs identified 40 unique glycopeptides. In addition, ETD-MS2 provided that a cleavage specificity by OperATOR was at the N-termini of the O-linked glycosylation sites with core 1 glycans. Therefore, one of the advantages of EXoO method for O-linked glycoproteomics analysis empowered efficient O-linked glycosylation site localization by high cleavage specificity of OperATOR that was confirmed by ETD-MS2.

6. The authors still claim that they have unambiguous site assignments. The concern about Ser/Thr found at positions +1 and +2 was not addressed appropriately. The simple fetuin model may not reflect the performance with the diversity of the entire proteome. Here, running the same sample in both HCD and ETD would give an answer.

Response: the "unambiguous" has been removed as described in response to reviewer's comment #1. Reviewer has concern about Ser/Thr found at positions +1 and +2. As suggested by reviewer, serum O-linked glycopeptides generated by EXoO was analyzed using sequential ETD/HCD-MS2. The analysis showed that the OperATOR has high cleavage specificity to yield site-specific glycopeptides with the O-glycosites at the first amino acid position of the glycopeptides. The analysis and data is in the new section: **Specificities of OperATOR for peptides and O-linked glycans** in p8 line 162-201.

7. *If the Operator specificity is unknown it may be premature to use it to explore aberrant glycoproteomes (p11) without discussion of the limitations - namely that increase in core1 O-glycans with sialic acid capping is likely the only property measured and interpretations of how this comes about may be difficult?*

Response: To address the reviewer's concern on OperATOR specificity, we investigated the two sources of study using this enzyme (AC paper and the company as described above in response to reviewer's comment #3). In addition, we have conducted ETD-MS2 to try to determine the cleavage specificity in the new result section as described above. For glycan specificity, we revised the manuscript to discuss the limitation of EXoO in the study of cancer tissue at p13 line 284-286: "Owing to unclear substrate specificity of OperATOR for O-linked glycans, the site-specific O-linked glycosylation by glycans in addition to Core 1 glycans merits future investigation." The specificity for sialic acid containing O-glycans is discussed in response to reviewer's comment #3.

8. *The final conclusion that "EXoO identified a substantially larger number of sites and glycoproteins almost doubling of the number of sites identified in decades" is incorrect and the authors need to survey the literature for recent reports using different lectin enrichments with cell lines, plasma and even organs. Finally, many citations are still random and in places completely out of scope.*

Response: We compared our data to the three major databases including O-GalNAc human SimpleCell glycoproteome DB (Steentoft et al, 2013; Steentoft et al, 2011), PhosphoSitePlus (Hornbeck et al, 2015) and Uniprot database (UniProt Consortium, 2018). We revised the abstract p2 line 30-32: "This large-scale localization of O-linked glycosylation sites demonstrated that EXoO is an effective method for defining the site-specific O-linked glycoproteome in different types of sample." And also p16 line 347-351: "EXoO identified a large number of O-linked glycosylation sites and glycoproteins with 2,580 novel O-linked glycosylation sites that are not reported in three major database including O-GalNAc human SimpleCell glycoproteome DB (Steentoft et al, 2013; Steentoft et al, 2011), PhosphoSitePlus (Hornbeck et al, 2015) and Uniprot database (UniProt Consortium, 2018)." We have revised the citations accordingly.

9. *Authors agree that only 69% of their data has glycan composition to be clear associated with one structure while the rest 31% cases glyco site numebrs could be over estimated. On the other hand the total number of glycosites still the same (3000).*

Response: the reviewer has concern regarding the 31% cases of site-specific O-glycopeptides identified by EXoO that might have more than one O-glycan structure and may be due to the multiple O-linked glycosylation sites presented in one glycopeptide.

In our previous study, HCD-MS2 has been shown to be able to identify peptide backbones of glycopeptides with different glycan compositions (Yang et al, 2014). On the other hand, the glycopeptides generate –b and –y ions in the spectra in HCD mode regardless of different glycan composition that leads to the observation that similar spectra are generated for glycopeptides with different glycan composition. See also figure below:

Figure 3. MS/MS spectra of O-linked glycopeptides demonstrating the applicability of spectral-aligning strategy for identification of O-linked glycopeptides. Peptide IEPLGVAPT⁴⁹⁹KAK in different glycoforms Gal₁GalNAc₁ in (A) and NeuAc₁Gal₁GlcNAc₁GalNAc₁ in (B) are aligned to show a similar pattern of peaks.

This is because the glycan portion is fragmented, and the spectra contain primarily peptide -b and -y ions in the HCD fragmentation.

We observed the similarly results from data in this manuscript, O-glycopeptides with one O-glycan (upper spectrum in the figure below) or multiple O-glycans (middle spectrum in the figure below) generated similar HCD spectra. The upper spectrum in figure below identified O-glycopeptide TSAHGNVAEGETkPDPDVTER + HexHexNAc. The middle spectrum in figure below identified TSAHGNVAEGETkPDPDVTER + Hex(2)HexNAc(2) or 2xHexHexNAc that belonged to the 31% cases with more than one glycan. The ETD-MS analysis (lower spectrum in the figure below) confirmed the observation that there were two O-glycosites at T and S sites in T*S*AHGNVAEGETkPDPDVTER with ptmRS site probabilities of 100 for both of the O-glycosites. On the other hand, we showed that HCD-MS2 could efficiently identify peptide backbone of O-glycopeptides with different or multiple O-glycans. In the case of multiple O-glycans in the glycopeptides, more O-glycosites may present in the identified O-glycopeptides.

c and z ion chart for ETD-MS2 above

10. The authors responded that manual validation was performed, and false cases were filtered out. It is difficult to judge (very subjective and based on the user experience), but just as an example in Dataset EV1 cases with the Xcore value below 1 were found. It is especially strange because some of these cases were reported heavily glycosylated. For example peptide SAVPDAA...VGP with 6 monosaccharides per sequence with the Xcore 0,8. This is very hard to accept as a true identification.

Response: We manually inspected the MS/MS data for validation of data, but we did not manually filter out any data passing 1% FDR. As stated in our previous response to reviewers: "We have manually inspected few hundreds of identified glycopeptides with different scores using spectral viewer provided in the Proteome Discoverer 2.2." We agree with the reviewer's comment that it will be very subjective to make judge and filter out data. Filtering out data manually heavily depends on researchers' experience that will also be considered as "cherry picking".

Reviewer has concern on a peptide with 6 monosaccharides and an Xcorr score of 0.8. HCD-MS can identify peptide backbone with different number of monosaccharides as described in the above response. The 6 monosaccharides might consist of three core 1 HexHexNAc. Considering that the identified peptide S*AVPDAAAGPT*PS*AAGPPVAS*VVVGP has three known and one potential new O-glycosites marked in stars in peptide sequence, it is not impossible to see 6 monosaccharides. The Xcorr is one of the factors used in the FDR calculation in the SEQUEST search engine with Percolator. Other scores are also considered in the FDR

calculation. As the Xcorr 0.8 was the lowest score in all 193 PSMs assigned to fetuin from our study, this PSM may be considered in the grey area where will be hard to manually judge its correctness. Given that 1% FDR is used, the one PSM with Xcorr 0.8 in 192 PSMs with Xcorr over 1 is tolerated within 1% FDR.

11. This reviewer would recommend to include a supplemental annotated spectra. Proteome Discoverer has this function.

Response: as requested by reviewer, we have included supplemental annotated spectra in the appendix file. However, because we have thousands of assigned spectra in the data, we randomly pick 112 spectra (56 for Ser and 56 for Thr O-glycosites) cross the tumor and normal dataset to provide spectral examples with different peptide length, charge, sequence and scores. In the manuscript, we have revised to add these annotated spectra with description: "A number of 112 spectra with different sequences, charge, peptide length, scores and glycan compositions were annotated (Appendix Fig S2)" in p7 line 148-150.

3rd Editorial Decision

16th of October 2018

Thank you for sending us your revised manuscript. We have now heard back from reviewer #1 who was asked to evaluate your manuscript. As you will see below, s/he is satisfied with the performed revisions and is supportive of publication. S/he lists however a series of issues, mainly referring to text modifications, which we would ask you to address in a minor revision.

Moreover, before we formally accept the manuscript for publication, we would ask you to address a couple of remaining editorial issues listed below.

REFEREE REPORTS

Reviewer #1:

The revised manuscript essentially addresses all concerns and there is no need to review again, but for the authors consideration some responses and suggestions for improving the text and clarity is included below after each comment.

1. The numbers game with 3,000 sites and 1,000 proteins still used has to be revised given the ambiguities pointed out with the MS identifications. The statement p7 of unambiguously mapped sites does not seem to be correct?

Reponses: In the response below, we have provided experimental data and explanation to address each of the sources of the ambiguities (provided below in responses to the specific questions on this topic). Given the ambiguities with the MS identifications, we revised the manuscript and deleted the "unambiguously" in page 7 and other places in the manuscript.

Reviewer #1: Addressed, but the entire text authors use "sites" with the defined numbers, but this should be limited to unambiguous site assignments.

2. The Introduction is still highly biased and trying to lead up to a new method without reduction in sample complexity and with information of natural glycan structures. Since this is the holy grail of the field it is important upfront to bring the reader into the reality that the methods at best can report only core1 structures after removal of sialic acids. The authors still fail to discuss that current lectin enrichments and ETD MS of desialylated core1 glycoproteomes reach >1-2,000 O-glycosites in plasma and organs? See e.g. PMID: 29296958?

Response: we apologize for misunderstanding the reviewer may have. We did not try to lead up a new method without reduction in sample complexity. In fact, we needed to reduce sample

complexity by using SAX column to enrich intact O-linked glycopeptides. Next, the use of solid-phase support was needed to further capture peptides and reduce sample complexity via specific release of site-specific O-linked glycopeptides. We revised the introduction by replacing the sentence "All of these methodologies have reduced the sample complexity by enrichment of O-linked glycopeptides from background peptides, which remains as a severe hindrance to the structural and functional study of O-linked glycoproteins." with the sentence "The enrichment methodologies have proved to be useful to study O-linked glycoproteome in different biological systems." In p4 line 69-70.

Reviewer has concern regarding introduction that our method can report all natural glycan structures instead of reporting core 1 structures after removal of sialic acids. To address this concern, we revised the introduction by replacing the sentence "It has been designed to simultaneously enrich and identify O-linked glycosylation sites and define their site-specific glycans" with the sentence to "It has been designed to simultaneously enrich and identify O-linked glycosylation sites and define their site-specific glycans with primarily core 1 structures with or without sialic acids" in p5 line 98-100. The EXoO method can work on core 1 structures without removal of sialic acids. Please refer to response to reviewer's comment #3 below for the data and discussion regarding the use of sialidase.

Reviewer requests to further discuss PMID: 29296958 regarding lectin enrichment and ETD-MS analysis of glycoproteome in plasma and organs in PMID: 29296958. In our previous manuscript, we have cited this article with several others on the O-glycoproteomic methods using lectins. As requested by the reviewer, in this revision, we described the article, PMID: 29296958, in details. We have re-checked: Characterizing the O-glycosylation landscape of human plasma, platelets, and endothelial cells. In the abstract of the paper, "native tissue" has been mentioned. However, the paper only studied plasma, platelets and endothelial cells mentioned in the title and described in the Materials and Methods section: "AB RhD-positive platelets from 4 random donors and plasma were obtained from the Blood Bank of the Capital Region and harvested according to standard protocols. Primary Human Umbilical Vein Endothelial Cells (HUVEC) were purchased from Life Technologies." The HUVEC is cells cultured in medium 200PRF supplemented with low serum growth supplement as described in the Supplement Methods of PMID: 29296958. Therefore, there is no data for organ or tissue in the study described in PMID: 29296958.

Regarding the plasma data described in article PMID: 29296958, the authors described "the hitherto largest O-glycoproteome", among the total number of 1,123 O-glycosites reported in this study, 354 O-glycosites were identified from plasma. We therefore revised the manuscript introduction in p4 line 77-80 to describe the study: "In the largest O-glycoproteome study, lectin enrichments of desialylated core 1 O-linked glycopeptides from plasma using PNA and VVA and ETD-MS2 analysis, 354 unique O-linked glycosylation sites were reported among the total of 1,123 O-glycosites identified from plasma, platelets, and endothelial cells (King et al, 2017)."

Reviewer #1: Partially addressed. Comment about reduction of the sample complexity was clarified but statements are still unclear. For example on p.5 line 98-100 the authors state "It has been designed to simultaneously....with primarily core 1 structures with and without sialic acid". From ETD data (EV5) applied in parallel for the analysis of human serum only core1 structures were confirmed. It is recommended to remove "primarily" while this bring to the misunderstanding that the other core structures have been identified with the less extend. Also authors say with and without sialic acids, but in all result tables (EV1-Ev5) only neutral structures are reported.

3. Furthermore, the Introduction should discuss the Anal Chem paper with specific information (not currently without specifics at the end of the Discussion) and fully disclose available information of the substrate specificity of the Operator enzyme.

Response: we agree with reviewer. Therefore, we have surveyed the data produced from the company and also in the Yang et al. Anal Chem 2018.

Specifically, information from the company shows that OpeRATOR is active on sialylated O-glycoproteins (lane 2 in figure below). In lane 2, TNF α receptor can be digested by OpeRATOR without sialidase treatment. Lane 3 shows that the TNF α receptor contain sialic acids and treatment of sialidase removed sialic acids from the protein and decreased the molecular weight.

In addition, information from poster from the company shows that protein can be digested with OpeRATOR without sialidase (red square in the workflow below) that leads to identification of a glycopeptide S.TSFLPMGP.S with HexHexNAcNeuAc (red square in the table below). The figure below is cut and recognized from company's poster.

Original section from the poster:

Moreover, in data from Yang et al. Anal Chem 2018 where bovine submaxillary gland mucin (MBS) O-glycosites are studied. As the authors stated that "We compared the identification of O-glycosites with and without sialic acids by treating with neuraminidase (Supporting Information Table S3)." We checked their supporting information table S3 and filtered to display glycopeptides with sialic acid (table below). Their data appeared to show glycans with NeuAc and/or NeuGc. For example, glycopeptide K.tSQQL.S with Hex(1)HexNAc(1)NeuAc(1). We also see L.sVR.V with Hex(2)HexNAc(2)NeuAc(2). Given no data analysis section in their paper, we are not confident to comment that the OpeRATOR can recognize core 2 structures because sVR contains only one O-glycosite in the sequence and the glycan composition can only be interpreted to be a core 2 structure.

In the previous study published by AC, the cleavage specificity of OpeRATOR was not examined. As requested by this reviewer, we revised the manuscript to discuss the Anal Chem paper in the introduction in p5 line 107-112: "During the peer-review period of our study, a manuscript by Yang et al. (Yang et al, 2018), described the analysis of O-linked glycosylation sites from several simple glycoproteins including fetuin, mucin, and Zika viral proteins. This study took use of the enzyme, Operator, with cleavage of peptide sequences at the N-termini of the O-linked glycosylation sites (Yang et al, 2018). However, the glycan specificity and cleavage specificity for the O-linked glycosylation sites by Operator enzyme is not clearly defined."

The reviewer asks to fully disclose available information of the substrate specificity of the Operator enzyme. With our study described in the original manuscript using HCD-MS2 and the additional ETD-MS2 data described in the revised manuscript, we have disclosed the enzymatic specificity information supported by data presented in this study and to the best of our knowledge.

Reviewer #1: Addressed, but since the authors introduced new ETD MS2 data a new consideration is relevant: Tables (EV1-EV5) report glycan PTM as a cumulative compositions (Hex(1)HexNAc(1) or Hex(2)HexNAc(2)) but in the "Peptide Modification" column (for example serum data in eV4) it reads T1(Hex(2)HexNAc(2)), which indicates a Core2 structure on T1 position. However, the ETD data of the same sample clearly confirm that this is two Core1 Hex(1)HexNAc(1) structures on 1 and T2 (or even S5), which stresses the point what is the actual site numbers and what kind of glyco structures are reported. This is still confusing to the reader.

4. The statement "sialidase could be omitted if the focus is to define site specific glycans with sialic acids", needs further qualification? The producer recommends use of sialidase presumably due to efficiency reasons, but without data this could at best only be a hypothesis that needs to be tested?

Response: The above experimental data in response to the reviewer's comment #3 support that the O-linked glycosylation sites containing sialic acids O-glycans can be released by OpeRATOR. However, the company also recommends removing the sialic acids to increase the efficiency of cleavage by OpeRATOR.

Therefore, we have revised the manuscript and credited the original recommendation by the company. In p15 324-327: "As stated by manufacture, the use of sialidase in the procedure facilitated efficient cleavage by OpeRATOR that was used to improve mapping of O-linked glycosylation sites. The addition of sialidase could be omitted if the study focus is to define site-specific glycan structures with sialic acid."

Reviewer #1. Addressed, but same comment as above.

5. As commented previously the discussion of ETD vs HCD needs to be balanced and include pitfalls with HCD. Without direct comparison (as suggested) the conclusions drawn may not be true. This reviewer's experience is that current MS with OrbiTraps provide similar identification numbers for glycopeptides in both ETD and HCD modes, although the datasets may vary. It is correct that for ETD MS/MS precursor ions has to be multiply charged and the fragmentation pattern is better when the charge is more than 3. However, precursor ions with z=2+ still provide decent fragmentation patterns and in our experience the majority of glycopeptides have charge states of 3 or 4 in any case. Thus, the key issue of unambiguous assignments with HCD needs to be discussed.

Response: we apologize for the confusion, but we did not compare and discuss ETD vs HCD in the previous revision of the manuscript.

In study of tryptic glycopeptides, while ETD and HCD may identify similar number for glycopeptides. The site localization will have to rely on ETD data. HCD cannot confidently assign

O-glycosites in the glycopeptides due to preferential fragmentation of glycans in the HCD mode. To reflect this fact, we revised the manuscript in the introduction to state in p4 80-83: "Alternative to ETD fragmentation, HCD-MS2 may provide an efficient fragmentation method to identify glycopeptides but the O-linked glycosylation sites cannot be confidently assigned due to preferential fragmentation of O-linked glycans during HCD mode (Yang et al, 2014)."

Prior to this study, there is no ETD data on the type of glycopeptides cleaved by OpeRATOR. Site-specific O-glycopeptides generated by trypsin and OpeRATOR can be shorter than tryptic glycopeptides that many of the precursors have +2 charge. As an experimental evidence, we examined our data and found that 38% PSMs in the tissue data are glycopeptides with +2 charges. The precursor with +2 charges may not be efficiently fragmented in the ETD mode. To have a discussion of ETD vs HCD on glycopeptides generated by OpeRATOR as requested by the reviewer, we conducted additional experiment to use sequential ETD/HCD and wrote an addition result section in p8 line 162-201: Specificities of OpeRATOR for peptides and O-linked glycans. In this additional result section and the addition dataset EV5, we found that HCD-MS2 identified 85 unique glycopeptides and ETD-MSs identified 40 unique glycopeptides. In addition, ETD-MS2 provided that a cleavage specificity by OpeRATOR was at the N-termini of the O-linked glycosylation sites with core 1 glycans. Therefore, one of the advantages of EXoO method for O-linked glycoproteomics analysis empowered efficient O-linked glycosylation site localization by high cleavage specificity of OpeRATOR that was confirmed by ETD-MS2.

Reviewer #1: Addressed. This additional data exactly point to site ambiguity assignments. The authors should make it clear in the entire text and clarify in tables (EV1-EV5)

6. The authors still claim that they have unambiguous site assignments.

The concern about Ser/Thr found at positions +1 and +2 was not addressed appropriately. The simple fetuin model may not reflect the performance with the diversity of the entire proteome. Here, running the same sample in both HCD and ETD would give an answer.

Response: the "unambiguous" has been removed as described in response to reviewer's comment #1.

Reviewer has concern about Ser/Thr found at positions +1 and +2. As suggested by reviewer, serum O-linked glycopeptides generated by EXoO was analyzed using sequential ETD/HCD-MS2. The analysis showed that the OpeRATOR has high cleavage specificity to yield site-specific glycopeptides with the O-glycosites at the first amino acid position of the glycopeptides. The analysis and data is in the new section: Specificities of OpeRATOR for peptides and O-linked glycans in p8 line 162-201.

#Reviewer #1: Addressed.

7. If the Operator specificity is unknown it may be premature to use it to explore aberrant glycoproteomes (p11) without discussion of the limitations - namely that increase in core1 O-glycans with sialic acid capping is likely the only property measured and interpretations of how this comes about may be difficult?

Response: To address the reviewer's concern on OpeRATOR specificity, we investigated the two sources of study using this enzyme (AC paper and the company as described above in response to reviewer's comment #3). In addition, we have conducted ETD-MS2 to try to determine the cleavage specificity in the new result section as described above. For glycan specificity, we revised the manuscript to discuss the limitation of EXoO in the study of cancer tissue at p13 line 284-286: "Owing to unclear substrate specificity of OpeRATOR for O-linked glycans, the site-specific O-linked glycosylation by glycans in addition to Core 1 glycans merits future investigation." The specificity for sialic acid containing O-glycans is discussed in response to reviewer's comment #3. Reviewer #1: Addressed

8. The final conclusion that "EXoO identified a substantially larger number of sites and glycoproteins almost doubling of the number of sites identified in decades" is incorrect and the authors need to survey the literature for recent reports using different lectin enrichments with cell lines, plasma and even organs. Finally, many citations are still random and in places completely out of scope.

Response: We compared our data to the three major databases including O-GalNAc human SimpleCell glycoproteome DB (Steentoft et al, 2013; Steentoft et al, 2011), PhosphoSitePlus (Hornbeck et al, 2015) and Uniprot database (UniProt Consortium, 2018). We revised the abstract p2 line 30-32: "This large-scale localization of O-linked glycosylation sites demonstrated that EXoO is an effective method for defining the site-specific O-linked glycoproteome in different types of sample." And also p16 line 347-351: "EXoO identified a large number of O-linked glycosylation sites and glycoproteins with 2,580 novel O-linked glycosylation sites that are not reported in three major database including O-GalNAc human SimpleCell glycoproteome DB (Steentoft et al, 2013;

Steenfot et al, 2011), PhosphoSitePlus (Hornbeck et al, 2015) and Uniprot database (UniProt Consortium, 2018)." We have revised the citations accordingly.

Reviewer #1: Partially addressed. PhosphoSite Plus is not a unique DB but contains data from two other DB as mentioned by the authors. Why did the authors not use data from the resources mentioned in introduction (PMID: 29296958, Darula&Medziradski, Bard, etc)? There are also a number of other groups reporting O-glycoproteomics of tissue and serum.

9. Authors agree that only 69% of their data has glycan composition to be clear associated with one structure while the rest 31% cases glyco site numebtrs could be over estimated. On the other hand the total number of glycosites still the same (3000).

Response: the reviewer has concern regarding the 31% cases of site-specific O-glycopeptides identified by EXoO that might have more than one O-glycan structure and may be due to the multiple O-linked glycosylation sites presented in one glycopeptide.

In our previous study, HCD-MS2 has been shown to be able to identify peptide backbones of glycopeptides with different glycan compositions (Yang et al, 2014). On the other hand, the glycopeptides generate -b and -y ions in the spectra in HCD mode regardless of different glycan composition that leads to the observation that similar spectra are generated for glycopeptides with different glycan composition. See also figure below:

This is because the glycan portion is fragmented, and the spectra contain primarily peptide -b and -y ions in the HCD fragmentation.

We observed the similarly results from data in this manuscript, O-glycopeptides with one O-glycan (upper spectrum in the figure below) or multiple O-glycans (middle spectrum in the figure below) generated similar HCD spectra. The upper spectrum in figure below identified O-glycopeptide TSAHGNVAEGETkPDPDVTER + HexHexNAc. The middle spectrum in figure below identified TSAHGNVAEGETkPDPDVTER + Hex(2)HexNAc(2) or 2xHexHexNAc that belonged to the 31% cases with more than one glycan. The ETD-MS analysis (lower spectrum in the figure below) confirmed the observation that there were two O-glycosites at T and S sites in T*S*A*HGNVAEGETkPDPDVTER with ptmRS site probabilities of 100 for both of the O-glycosites. On the other hand, we showed that HCD-MS2 could efficiently identify peptide backbone of O-glycopeptides with different or multiple O-glycans. In the case of multiple O-glycans in the glycopeptides, more O-glycosites may present in the identified O-glycopeptides.

Reviewer #1: Addressed.

10. The authors responded that manual validation was performed, and false cases were filtered out. It is difficult to judge (very subjective and based on the user experience), but just as an example in Dataset EV1 cases with the Xcore value below 1 were found. It is especially strange because some of these cases were reported heavily glycosylated. For example peptide SAVPDAA....VGP with 6 monosaccharides per sequence with the Xcore 0,8. This is very hard to accept as a true identification.

Response: We manually inspected the MS/MS data for validation of data, but we did not manually filter out any data passing 1% FDR. As stated in our previous response to reviewers: "We have manually inspected few hundreds of identified glycopeptides with different scores using spectral viewer provided in the Proteome Discoverer 2.2." We agree with the reviewer's comment that it will be very subjective to make judge and filter out data. Filtering out data manually heavily depends on researchers' experience that will also be considered as "cherry picking".

Reviewer has concern on a peptide with 6 monosaccharides and an Xcorr score of 0.8. HCD-MS can identify peptide backbone with different number of monosaccharides as described in the above response. The 6 monosaccharides might consist of three core 1 HexHexNAc. Considering that the identified peptide S*AVPDAAGPT*PS*AAGPPVAS*VVVGP has three known and one potential new O-glycosites marked in stars in peptide sequence, it is not impossible to see 6 monosaccharides. The Xcorr is one of the factors used in the FDR calculation in the SEQUEST search engine with Percolator. Other scores are also considered in the FDR calculation. As the Xcorr 0.8 was the lowest score in all 193 PSMs assigned to fetuin from our study, this PSM may be considered in the grey area where will be hard to manually judge its correctness. Given that 1% FDR is used, the one PSM with Xcorr 0.8 in 192 PSMs with Xcorr over 1 is tolerated within 1% FDR.

Reviewer #1: Partially addressed. Agree that 1% FDR is a criteria in proteomics, however, there is no systematic study for how to define FDR for glycoproteomics. Especially in the case of Perqolator a number of low score hits that are accepted within the 1% FDR may actually upon manual inspection show poor fragmentation patterns. It may be advisable to use Xcore cut off criteria as

well, especially with Perqolator.

11. This reviewer would recommend to include a supplemental annotated spectra. Proteome Discoverer has this function.

Response: as requested by reviewer, we have included supplemental annotated spectra in the appendix file. However, because we have thousands of assigned spectra in the data, we randomly pick 112 spectra (56 for Ser and 56 for Thr O-glycosites) cross the tumor and normal dataset to provide spectral examples with different peptide length, charge, sequence and scores. In the manuscript, we have revised to add these annotated spectra with description: "A number of 112 spectra with different sequences, charge, peptide length, scores and glycan compositions were annotated (Appendix Fig S2)" in p7 line 148-150.

Reviewer #1: Addressed.

3rd Revision - authors' response

16th of October 2018

Response to reviewer's comments:

Reviewer #1:

The revised manuscript essentially addresses all concerns and there is no need to review again, but for the authors consideration some responses and suggestions for improving the text and clarity is included below after each comment.

1. The numbers game with 3,000 sites and 1,000 proteins still used has to be revised given the ambiguities pointed out with the MS identifications. The statement p7 of unambiguously mapped sites does not seem to be correct?

Reponses: In the response below, we have provided experimental data and explanation to address each of the sources of the ambiguities (provided below in responses to the specific questions on this topic). Given the ambiguities with the MS identifications, we revised the manuscript and deleted the "unambiguously" in page 7 and other places in the manuscript.

Reviewer #1: Addressed, but the entire text authors use "sites" with the defined numbers, but this should be limited to unambiguous site assignments.

2. The Introduction is still highly biased and trying to lead up to a new method without reduction in sample complexity and with information of natural glycan structures. Since this is the holy grail of the field it is important upfront to bring the reader into the reality that the methods at best can report only core1 structures after removal of sialic acids. The authors still fail to discuss that current lectin enrichments and ETD MS of desialylated core1 glycoproteomes reach >1-2,000 O-glycosites in plasma and organs? See e.g. PMID: 29296958?

Response: we apologize for misunderstanding the reviewer may have. We did not try to lead up a new method without reduction in sample complexity. In fact, we needed to reduce sample complexity by using SAX column to enrich intact O-linked glycopeptides. Next, the use of solid-phase support was needed to further capture peptides and reduce sample complexity via specific release of site-specific O-linked glycopeptides. We revised the introduction by replacing the sentence "All of these methodologies have reduced the sample complexity by enrichment of O-linked glycopeptides from background peptides, which remains as a severe hindrance to the structural and functional study of O-linked glycoproteins." with the sentence "The enrichment methodologies have proved to be useful to study O-linked glycoproteome in different biological systems." In p4 line 69-70.

Reviewer has concern regarding introduction that our method can report all natural glycan structures instead of reporting core 1 structures after removal of sialic acids. To address this concern, we revised the introduction by replacing the sentence "It has been designed to simultaneously enrich and identify O-linked glycosylation sites and define their site-specific glycans" with the sentence to "It has been designed to simultaneously enrich and identify O-linked glycosylation sites and define their site-specific glycans with primarily core 1 structures with or without sialic acids" in p5 line 98-100. The EXoO method can work on core 1 structures without removal of sialic acids. Please refer to response to reviewer's comment #3 below for the data and discussion regarding the use of sialidase.

Reviewer requests to further discuss PMID: 29296958 regarding lectin enrichment and ETD-MS analysis of glycoproteome in plasma and organs in PMID: 29296958. In our previous manuscript, we have cited this article with several others on the O-glycoproteomic methods using lectins. As requested by the reviewer, in this revision, we described the article, PMID: 29296958, in details. We have re-checked: Characterizing the O-glycosylation landscape of human plasma, platelets, and endothelial cells. In the abstract of the paper, "native tissue" has been mentioned. However, the paper only studied plasma, platelets and endothelial cells mentioned in the title and described in the Materials and Methods section: "AB RhD-positive platelets from 4 random donors and plasma were obtained from the Blood Bank of the Capital Region and harvested according to standard protocols. Primary Human Umbilical Vein Endothelial Cells (HUVEC) were purchased from Life Technologies." The HUVEC is cells cultured in medium 200PRF supplemented with low serum growth supplement as described in the Supplement Methods of PMID: 29296958. Therefore, there is no data for organ or tissue in the study described in PMID: 29296958.

Regarding the plasma data described in article PMID: 29296958, the authors described "the hitherto largest O-glycoproteome", among the total number of 1,123 O-glycosites reported in this study, 354 O-glycosites were identified from plasma. We therefore revised the manuscript introduction in p4 line 77-80 to describe the study: "In the largest O-glycoproteome study, lectin enrichments of de-sialylated core 1 O-linked glycopeptides from plasma using PNA and VVA and ETD-MS2 analysis, 354 unique O-linked glycosylation sites were reported among the total of 1,123 O-glycosites identified from plasma, platelets, and endothelial cells (King et al, 2017)."

Reviewer #1: **Partially addressed.** Comment about reduction of the sample complexity was clarified but statements are still unclear. For example on p.5 line 98-100 the authors state "It has been designed to simultaneously....with primarily core 1 structures with and without sialic acid". From ETD data (EV5) applied in parallel for the analysis of human serum only core1 structures were confirmed. It is recommended to remove "primarily" while this bring to the misunderstanding that the other core structures have been identified with the less extend. Also authors say with and without sialic acids, but in all result tables (EV1-Ev5) only neutral structures are reported.

Response: as suggested by reviewer, we have revised the manuscript to remove "primarily" on p.5 line 99.

The issue about sialic acids has been addressed in the response to reviewer's comment #3 below.

3. Furthermore, the Introduction should discuss the Anal Chem paper with specific information (not currently without specifics at the end of the Discussion) and fully disclose available information of the substrate specificity of the Operator enzyme.

Response: we agree with reviewer. Therefore, we have surveyed the data produced from the company and also in the Yang et al. Anal Chem 2018.

Specifically, information from the company shows that OpeRATOR is active on sialylated O-glycoproteins (lane 2 in figure below). In lane 2, TNF α receptor can be digested by OpeRATOR without sialidase treatment. Lane 3 shows that the TNF α receptor contain sialic acids and treatment of sialidase removed sialic acids from the protein and decreased the molecular weight.

In addition, information from poster from the company shows that protein can be digested with OpeRATOR without sialidase (red square in the workflow below) that leads to identification of a glycopeptide S.TSFLPMGP.S with HexHexNAcNeuAc (red square in the table below). The figure below is cut and recognized from company's poster.

Original section from the poster:

Moreover, in data from Yang et al. Anal Chem 2018 where bovine submaxillary gland mucin (MBS) O-glycosites are studied. As the authors stated that "We compared the identification of O-glycosites with and without sialic acids by treating with neuraminidase (Supporting Information Table S3)." We checked their supporting information table S3 and filtered to display glycopeptides with sialic acid (table below). Their data appeared to show glycans with NeuAc and/or NeuGc. For example, glycopeptide K.tSQQL.S with Hex(1)HexNAc(1)NeuAc(1). We also see L.sVR.V with Hex(2)HexNAc(2)NeuAc(2). Given no data analysis section in their paper, we are not confident to comment that the OpeRATOR can recognizes core 2 structures because sVR contains only one O-glycosite in the sequence and the glycan composition can only be interpreted to be a core 2 structure.

In the previous study published by AC, the cleavage specificity of OpeRATOR was not examined. As requested by this reviewer, we revised the manuscript to discuss the Anal Chem paper in the

introduction in p5 line 107-112: "During the peer-review period of our study, a manuscript by Yang et al, (Yang et al, 2018), described the analysis of O-linked glycosylation sites from several simple glycoproteins including fetuin, mucin, and Zika viral proteins. This study took use of the enzyme, Operator, with cleavage of peptide sequences at the N-termini of the O-linked glycosylation sites (Yang et al, 2018). However, the glycan specificity and cleavage specificity for the O-linked glycosylation sites by Operator enzyme is not clearly defined."

The reviewer asks to fully disclose available information of the substrate specificity of the Operator enzyme. With our study described in the original manuscript using HCD-MS2 and the additional ETD-MS2 data described in the revised manuscript, we have disclosed the enzymatic specificity information supported by data presented in this study and to the best of our knowledge.

Reviewer #1: Addressed, but since the authors introduced new ETD MS2 data a new consideration is relevant: Tables (EV1-EV5) report glycan PTM as a cumulative compositions (Hex(1)HexNAc(1) or Hex(2)HexNAc(2)) but in the "Peptide Modification" column (for example serum data in eV4) it reads T1(Hex(2)HexNAc(2)), which indicates a Core2 structure on T1 position. However, the ETD data of the same sample clearly confirm that this is two Core1 Hex(1)HexNAc(1) structures on I and T2 (or even S5), which stresses the point what is the actual site numbers and what kind of glyco structures are reported. This is still confusing to the reader.

4. The statement "sialidase could be omitted if the focus is to define site specific glycans with sialic acids", needs further qualification? The producer recommends use of sialidase presumably due to efficiency reasons, but without data this could at best only be a hypothesis that needs to be tested? Response: The above experimental data in response to the reviewer's comment #3 support that the O-linked glycosylation sites containing sialic acids O-glycans can be released by OpeRATOR. However, the company also recommends removing the sialic acids to increase the efficiency of cleavage by OpeRATOR.

Therefore, we have revised the manuscript and credited the original recommendation by the company. In p15 324-327: "As stated by manufacture, the use of sialidase in the procedure facilitated efficient cleavage by OpeRATOR that was used to improve mapping of O-linked glycosylation sites. The addition of sialidase could be omitted if the study focus is to define site-specific glycan structures with sialic acid."

Reviewer #1. Addressed, but same comment as above.

5. As commented previously the discussion of ETD vs HCD needs to be balanced and include pitfalls with HCD. Without direct comparison (as suggested) the conclusions drawn may not be true. This reviewer's experience is that current MS with OrbiTraps provide similar identification numbers for glycopeptides in both ETD and HCD modes, although the datasets may vary. It is correct that for ETD MS/MS precursor ions has to be multiply charged and the fragmentation pattern is better when the charge is more than 3. However, precursor ions with $z=2+$ still provide decent fragmentation patterns and in our experience the majority of glycopeptides have charge states of 3 or 4 in any case. Thus, the key issue of unambiguous assignments with HCD needs to be discussed.

Response: we apologize for the confusion, but we did not compare and discuss ETD vs HCD in the previous revision of the manuscript.

In study of tryptic glycopeptides, while ETD and HCD may identify similar number for glycopeptides. The site localization will have to rely on ETD data. HCD cannot confidently assign O-glycosites in the glycopeptides due to preferential fragmentation of glycans in the HCD mode. To reflect this fact, we revised the manuscript in the introduction to state in p4 80-83: "Alternative to ETD fragmentation, HCD-MS2 may provide an efficient fragmentation method to identify glycopeptides but the O-linked glycosylation sites cannot be confidently assigned due to preferential fragmentation of O-linked glycans during HCD mode (Yang et al, 2014)."

Prior to this study, there is no ETD data on the type of glycopeptides cleaved by OpeRATOR. Site-specific O-glycopeptides generated by trypsin and OpeRATOR can be shorter than tryptic glycopeptides that many of the precursors have +2 charge. As an experimental evidence, we examined our data and found that 38% PSMs in the tissue data are glycopeptides with +2 charges. The precursor with +2 charges may not be efficiently fragmented in the ETD mode. To have a discussion of ETD vs HCD on glycopeptides generated by OpeRATOR as requested by the reviewer, we conducted additional experiment to use sequential ETD/HCD and wrote an addition result section in p8 line 162-201: Specificities of OpeRATOR for peptides and O-linked glycans. In this additional result section and the addition dataset EV5, we found that HCD-MS2 identified 85 unique

glycopeptides and ETD-MSs identified 40 unique glycopeptides. In addition, ETD-MS2 provided that a cleavage specificity by OpeRATOR was at the N-termini of the O-linked glycosylation sites with core 1 glycans. Therefore, one of the advantages of EXoO method for O-linked glycoproteomics analysis empowered efficient O-linked glycosylation site localization by high cleavage specificity of OpeRATOR that was confirmed by ETD-MS2.

Reviewer #1: Addressed. This additional data exactly point to site ambiguity assignments. The authors should make it clear in the entire text and clarify in tables (EV1-EV5)

6. The authors still claim that they have unambiguous site assignments.

The concern about Ser/Thr found at positions +1 and +2 was not addressed appropriately. The simple fetuin model may not reflect the performance with the diversity of the entire proteome. Here, running the same sample in both HCD and ETD would give an answer.

Response: the "unambiguous" has been removed as described in response to reviewer's comment #1. Reviewer has concern about Ser/Thr found at positions +1 and +2. As suggested by reviewer, serum O-linked glycopeptides generated by EXoO was analyzed using sequential ETD/HCD-MS2. The analysis showed that the OpeRATOR has high cleavage specificity to yield site-specific glycopeptides with the O-glycosites at the first amino acid position of the glycopeptides. The analysis and data is in the new section: Specificities of OpeRATOR for peptides and O-linked glycans in p8 line 162-201.

#Reviewer #1: Addressed.

7. If the Operator specificity is unknown it may be premature to use it to explore aberrant glycoproteomes (p11) without discussion of the limitations - namely that increase in core 1 O-glycans with sialic acid capping is likely the only property measured and interpretations of how this comes about may be difficult?

Response: To address the reviewer's concern on OpeRATOR specificity, we investigated the two sources of study using this enzyme (AC paper and the company as described above in response to reviewer's comment #3). In addition, we have conducted ETD-MS2 to try to determine the cleavage specificity in the new result section as described above. For glycan specificity, we revised the manuscript to discuss the limitation of EXoO in the study of cancer tissue at p13 line 284-286: "Owing to unclear substrate specificity of OpeRATOR for O-linked glycans, the site-specific O-linked glycosylation by glycans in addition to Core 1 glycans merits future investigation." The specificity for sialic acid containing O-glycans is discussed in response to reviewer's comment #3.

Reviewer #1: Addressed

8. The final conclusion that "EXoO identified a substantially larger number of sites and glycoproteins almost doubling of the number of sites identified in decades" is incorrect and the authors need to survey the literature for recent reports using different lectin enrichments with cell lines, plasma and even organs. Finally, many citations are still random and in places completely out of scope.

Response: We compared our data to the three major databases including O-GalNAc human SimpleCell glycoproteome DB (Steentoft et al, 2013; Steentoft et al, 2011), PhosphoSitePlus (Hornbeck et al, 2015) and Uniprot database (UniProt Consortium, 2018). We revised the abstract p2 line 30-32: "This large-scale localization of O-linked glycosylation sites demonstrated that EXoO is an effective method for defining the site-specific O-linked glycoproteome in different types of sample." And also p16 line 347-351: "EXoO identified a large number of O-linked glycosylation sites and glycoproteins with 2,580 novel O-linked glycosylation sites that are not reported in three major database including O-GalNAc human SimpleCell glycoproteome DB (Steentoft et al, 2013; Steentoft et al, 2011), PhosphoSitePlus (Hornbeck et al, 2015) and Uniprot database (UniProt Consortium, 2018)." We have revised the citations accordingly.

Reviewer #1: Partially addressed. PhosphoSite Plus is not a unique DB but contains data from two other DB as mentioned by the authors. Why did the authors not use data from the resources mentioned in introduction (PMID: 29296958, Darula&Medziradski, Bard, etc)? There are also a number of other groups reporting O-glycoproteomics of tissue and serum.

Response: the use of the three major O-glycoproteome database is because that they contain large number O-linked glycosylation sites collected from different studies. We did not have the intention to collect all the O-linked glycosylation sites from literatures.

9. Authors agree that only 69% of their data has glycan composition to be clear associated with one structure while the rest 31% cases glyco site numebrs could be over estimated. On the other hand the total number of glycosites still the same (3000).

Response: the reviewer has concern regarding the 31% cases of site-specific O-glycopeptides identified by EXoO that might have more than one O-glycan structure and may be due to the multiple O-linked glycosylation sites presented in one glycopeptide.

In our previous study, HCD-MS2 has been shown to be able to identify peptide backbones of glycopeptides with different glycan compositions (Yang et al, 2014). On the other hand, the glycopeptides generate -b and -y ions in the spectra in HCD mode regardless of different glycan composition that leads to the observation that similar spectra are generated for glycopeptides with different glycan composition. See also figure below:

This is because the glycan portion is fragmented, and the spectra contain primarily peptide -b and -y ions in the HCD fragmentation.

We observed the similarly results from data in this manuscript, O-glycopeptides with one O-glycan (upper spectrum in the figure below) or multiple O-glycans (middle spectrum in the figure below) generated similar HCD spectra. The upper spectrum in figure below identified O-glycopeptide TSAHGNVAEGETkPDPDVTER + HexHexNAc. The middle spectrum in figure below identified TSAHGNVAEGETkPDPDVTER + Hex(2)HexNAc(2) or 2xHexHexNAc that belonged to the 31% cases with more than one glycan. The ETD-MS analysis (lower spectrum in the figure below) confirmed the observation that there were two O-glycosites at T and S sites in T*S*AHG NVAEGETkPDPDVTER with ptmRS site probabilities of 100 for both of the O-glycosites. On the other hand, we showed that HCD-MS2 could efficiently identify peptide backbone of O-glycopeptides with different or multiple O-glycans. In the case of multiple O-glycans in the glycopeptides, more O-glycosites may present in the identified O-glycopeptides.

Reviewer #1: Addressed.

10. The authors responded that manual validation was performed, and false cases were filtered out. It is difficult to judge (very subjective and based on the user experience), but just as an example in Dataset EV1 cases with the Xcore value below 1 were found. It is especially strange because some of these cases were reported heavily glycosylated. For example peptide SAVPDAA...VGP with 6 monosaccharides per sequence with the Xcore 0,8. This is very hard to accept as a true identification.

Response: We manually inspected the MS/MS data for validation of data, but we did not manually filter out any data passing 1% FDR. As stated in our previous response to reviewers: "We have manually inspected few hundreds of identified glycopeptides with different scores using spectral viewer provided in the Proteome Discoverer 2.2." We agree with the reviewer's comment that it will be very subjective to make judge and filter out data. Filtering out data manually heavily depends on researchers' experience that will also be considered as "cherry picking".

Reviewer has concern on a peptide with 6 monosaccharides and an Xcorr score of 0.8. HCD-MS can identify peptide backbone with different number of monosaccharides as described in the above response. The 6 monosaccharides might consist of three core 1 HexHexNAc. Considering that the identified peptide S*AVPDAAAGPT*PS*AAGPPVAS*VVVGP has three known and one potential new O-glycosites marked in stars in peptide sequence, it is not impossible to see 6 monosaccharides. The Xcorr is one of the factors used in the FDR calculation in the SEQUEST search engine with Percolator. Other scores are also considered in the FDR calculation. As the Xcorr 0.8 was the lowest score in all 193 PSMs assigned to fetuin from our study, this PSM may be considered in the grey area where will be hard to manually judge its correctness. Given that 1% FDR is used, the one PSM with Xcorr 0.8 in 192 PSMs with Xcorr over 1 is tolerated within 1% FDR.

Reviewer #1: **Partially addressed.** Agree that 1% FDR is a criteria in proteomics, however, there is no systematic study for how to define FDR for glycoproteomics. Especially in the case of Perqolator a number of low score hits that are accepted within the 1% FDR may actually upon manual inspection show poor fragmentation patterns. It may be advisable to use Xcore cut off criteria as well, especially with Perqolator.

Response: we agree with reviewer that Xcorr may be used as a cut off criteria. However, a way to determinate the minimal Xcorr for O-glycoproteomics in different study settings will need future investigation.

11. This reviewer would recommend to include a supplemental annotated spectra. Proteome Discoverer has this function.

Response: as requested by reviewer, we have included supplemental annotated spectra in the appendix file. However, because we have thousands of assigned spectra in the data, we randomly pick 112 spectra (56 for Ser and 56 for Thr O-glycosites) cross the tumor and normal dataset to provide spectral examples with different peptide length, charge, sequence and scores. In the manuscript, we have revised to add these annotated spectra with description: "A number of 112 spectra with different sequences, charge, peptide length, scores and glycan compositions were annotated (Appendix Fig S2)" in p7 line 148-150.

Reviewer #1: Addressed.

Accepted

17th of October 2018

Thank you again for sending us your revised manuscript. We are now satisfied with the modifications made and I am pleased to inform you that your paper has been accepted for publication.

Corresponding Author Name: Hui Zhang
 Journal Submitted to: Molecular Systems Biology
 Manuscript Number: MSB-18-8486RR